# Aspirin's effect on kinetic parameters of cells contributes to its role in reducing incidence of advanced colorectal adenomas, shown by a multiscale computational study

Yifan Wang[1], C Richard Boland[2], Ajay Goel[3], Dominik Wodarz[1,4], Natalia L Komarova[1]*

[1]Department of Mathematics, University of California Irvine, Irvine, United States; [2]Department of Medicine, University of California San Diego School of Medicine, San Diego, United States; [3]Department of Molecular Diagnostics and Experimental Therapeutics, Beckman Research Institute of City of Hope Comprehensive Cancer Center, Duarte, United States; [4]Department of Population Health and Disease Prevention, University of California Irvine, Irvine, United States

*For correspondence: komarova@uci.edu

**Abstract** Aspirin intake has been shown to lead to significant protection against colorectal cancer, for example with an up to twofold reduction in colorectal adenoma incidence rates at higher doses. The mechanisms contributing to protection are not yet fully understood. While aspirin is an anti-inflammatory drug and can thus influence the tumor microenvironment, in vitro and in vivo experiments have recently shown that aspirin can also have a direct effect on cellular kinetics and fitness. It reduces the rate of tumor cell division and increases the rate of cell death. The question arises whether such changes in cellular fitness are sufficient to significantly contribute to the epidemiologically observed protection. To investigate this, we constructed a class of mathematical models of in vivo evolution of advanced adenomas, parameterized it with available estimates, and calculated population level incidence. Fitting the predictions to age incidence data revealed that only a model that included colonic crypt competition can account for the observed age-incidence curve. This model was then used to predict modified incidence patterns if cellular kinetics were altered as a result of aspirin treatment. We found that changes in cellular fitness that were within the experimentally observed ranges could reduce advanced adenoma incidence by a sufficient amount to account for age incidence data in aspirin-treated patient cohorts. While the mechanisms that contribute to the protective effect of aspirin are likely complex and multi-factorial, our study demonstrates that direct aspirin-induced changes of tumor cell fitness can significantly contribute to epidemiologically observed reduced incidence patterns.

## Editor's evaluation

This work develops a multistage/component mathematical model to analyze advanced colorectal adenomas and the impact that aspirin therapy has on adenoma formation rates. This study will be interesting to the cancer evolution community and in particular those interested in colorectal cancer incidence. While the model is mainly focused on aspirin chemoprevention, the model could be adapted to test other putative preventative agents, and thus could have a broad impact.

## Introduction

Colorectal cancer currently affects about 5% of the population in the USA and is a major cause of cancer-related deaths (*Siegel et al., 2020*). Prevention of colorectal cancer is an important goal in the quest to reduce morbidity and mortality. In this respect, long-term aspirin use has been shown to be effective (*Chan et al., 2012*; *Thun et al., 1991*). Aspirin is a non-steroidal anti-inflammatory drug (NSAID) and is a cyclo-oxygenase (COX)–2 inhibitor (*Goel et al., 2003*). The CAPP2 trial *Burn et al., 2011* demonstrated that the intake of 600 mg of aspirin per day for 2 years resulted in a 63% reduction in colorectal cancer incidence in Lynch Syndrome patients. Interestingly, observation of the protective effect of aspirin required a follow-up time of more than 55 months (*Burn et al., 2011*). In a range of studies, aspirin has also been shown to reduce incidence of sporadic colorectal cancer (*Thun et al., 1991*; *Tougeron et al., 2014*; *Friis et al., 2015*; *Lochhead and Chan, 2016*; *Chan et al., 2007*; *Drew et al., 2016*; *Chan et al., 2005*; *Rothwell, 2013*), and of adenomas (*Sandler et al., 2003*; *Chan et al., 2004*), which are a precursor of cancer. This was evident both in observational studies and in randomized controlled trials, which are reviewed for example in *Lochhead and Chan, 2016*; *Drew et al., 2016*. These studies report a reduction in cancer or adenoma incidence of the order of 10–50% in aspirin-treated compared to placebo groups, depending on the exact dose and frequency of aspirin intake. While some studies failed to detect significant protective effects of aspirin, larger studies with higher aspirin doses and longer treatment times yielded significant results.

The mechanisms underlying the protective effect of aspirin are likely complex and multi-factorial. Inflammation is a possible driver of colorectal carcinogenesis (*Itzkowitz and Yio, 2004*), and aspirin can reduce the extent of inflammation in the cellular microenvironment, which might contribute to a reduced development of disease. Other microenvironmental effects, such as the composition of the colorectal microbiome (*Prizment et al., 2020*; *Brennan et al., 2021*; *Zhao et al., 2020*), have also been shown to determine the degree of protection provided by aspirin. Our previous in vitro and in vivo work, however, has shown that physiologically relevant aspirin concentrations can have a direct effect on tumor cells, reducing their rate of proliferation and increasing their death rate (*Shimura et al., 2020*; *Zumwalt et al., 2017*). This not only results in reduced tumor growth, but can also lead to a lower probability that newly generated tumor cells successfully give rise to clonal expansion, thus increasing the likelihood that these initially transformed cells go extinct (*Wodarz et al., 2017*). This effect might contribute to the reduced incidence of colorectal cancer as a result of aspirin intake.

While these direct effects of aspirin on tumor cell division and death rates have been documented by us in vitro and in vivo (*Shimura et al., 2020*; *Zumwalt et al., 2017*), and occurred under physiologically realistic doses, it is unclear to what extent these changes in cellular kinetics can potentially alter disease incidence. To evaluate this quantitatively, a mathematical modeling framework needs to be developed that predicts epidemiological incidence data based on cellular processes. There is a rich history of such approaches in the cancer literature in different contexts (*Fisher and Hollomon, 1951*; *Nordling, 1953*; *Armitage and Doll, 1954*; *Luebeck and Moolgavkar, 2002*; *Meza et al., 2008*; *Moolgavkar, 1978*; *Hornsby et al., 2007*), which has allowed researchers to gain fundamental insights into carcinogenic processes based on the interpretation of age-incidence data. Here, we describe a mathematical model of advanced adenoma formation and parameterize it by fitting epidemiological predictions to incidence data that document advanced adenoma occurrence as a function of age. We then use this model to test whether aspirin-mediated changes in cellular kinetics, as documented by our experiments, can result in reductions in advanced adenoma incidence that are comparable to those observed in aspirin-treated patient cohorts. We find that the magnitude of changes in the kinetics of transformed cell populations that we observed experimentally can result in a pronounced reduction of advanced adenoma incidence, and that the epidemiologically observed incidence reductions (between 10% and 50%) can be explained by our model. This indicates that the direct effects of aspirin on dividing cells can in principle explain a significant amount of the chemoprotective effect exerted by this drug. We note, however, that while this is a clear result that emerges from this mathematical modeling effort, other mechanisms of aspirin not included in this model (such as anti-inflammatory effects [*Sostres et al., 2014*] or modulation of the microbiome [*Prizment et al., 2020*; *Brennan et al., 2021*; *Zhao et al., 2020*]) are likely to also contribute to the observed protective effect.

We start by describing a mathematical model of advanced adenoma formation and show that when parameterized with experimentally obtained estimates, it can account for epidemiologically observed

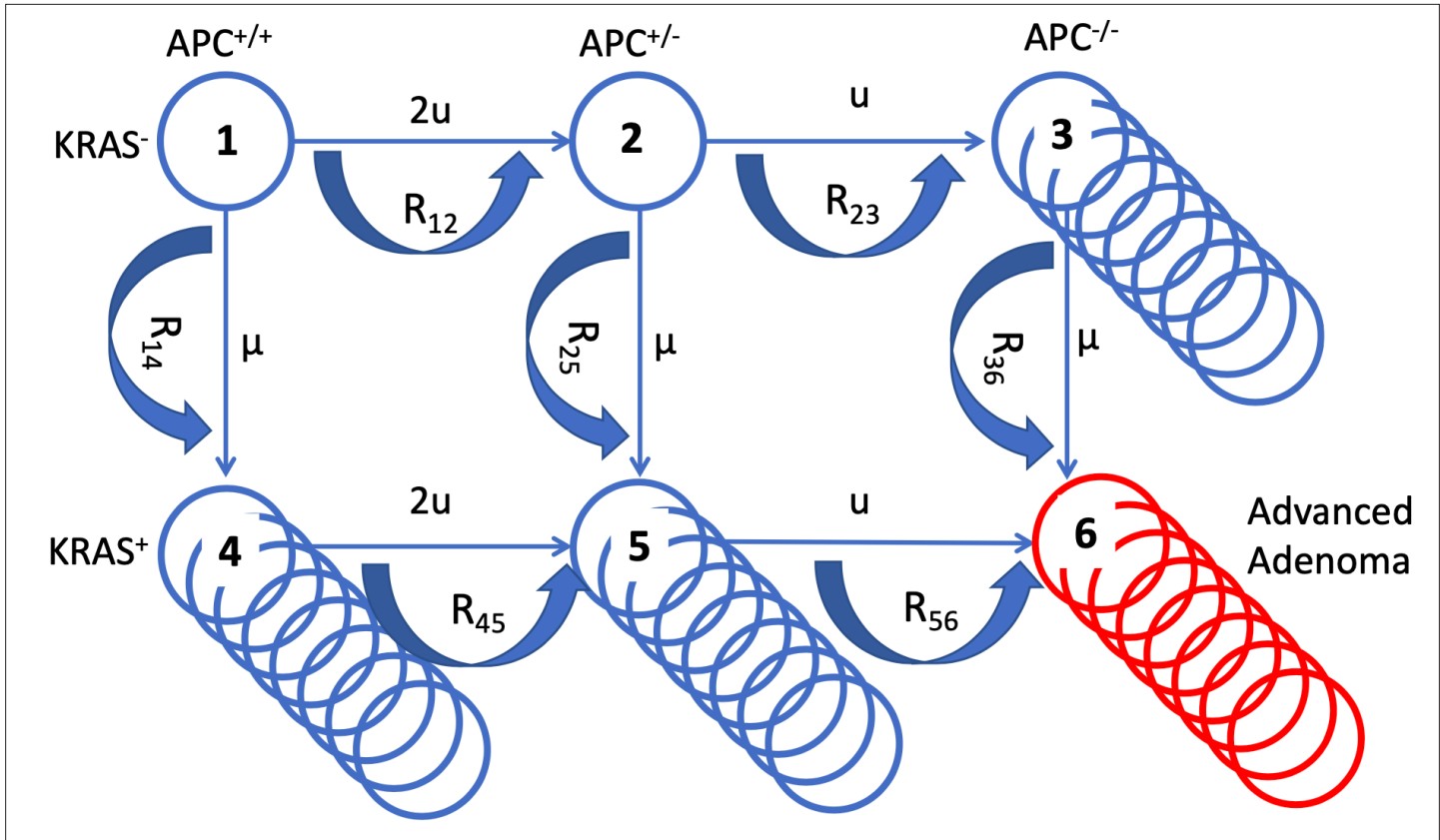

**Figure 1.** A schematic illustrating the mathematical model. The six cell types are denoted by circles, the mutation rates that give rise to different types are marked by the straight arrows. Type 6 (late adenoma) is marked in red. Crypt conversion rates are indicated by circular arrows and crypt fission by multiple circles.

## Results

### Computational modeling

In order to quantify the effects of aspirin on colorectal cancer initiation and progression, we have designed a mathematical model that is rooted in the process of multistep carcinogenesis (*Luebeck and Moolgavkar, 2002*; *Hornsby et al., 2007*; *Fearon, 2011*; *Ashley, 1969*). Its assumptions are similar in principle to those in a recent study (*Paterson et al., 2020*), with important differences that are discussed below. There are two early molecular events that we postulate (without assuming their temporal order): (1) An inactivation of the APC gene, or a related event that affects the functioning of the beta-catenin/WNT signaling pathway, and (2) an activation of the KRAS oncogene (or another gain-of-function mutation). For simplicity, we will be referring to these mutations as APC- and KRAS-mutations, keeping in mind that the model still applies in the presence of a pair of another loss-of-function and a gain-of-function mutation (further discussed below). The inactivation of the APC tumor suppressor gene is a classic example of a loss-of-function mutation, which implies two molecular events, corresponding to the inactivation of the two copies of the gene. The associated mutation rate is therefore assumed to be $u = 10^{-7}$ per cell division. The activation of the KRAS oncogene, on the other hand, is a gain-of-function event, whose mutation rate is about two orders of magnitude lower ($\mu = 10^{-9}$ per cell division). The associated selection-mutation diagram is shown in *Figure 1* and contains six different cell populations, denoted as types 1 through 6. The populations occupying the top row (types 1–3) are characterized by an unmutated KRAS oncogene; the populations

of the bottom row (types 4–6) all have the KRAS mutation activated. Moving from left to right on this diagram, the number of inactivated copies of the APC gene increases from 0 to 2, such that populations of types 1 and 4 are APC+/+, populations of types 2 and 5 are both APC+/-, and populations of types 3 and 6 are APC-/-.

These different populations correspond to different stages in the pathway towards colorectal cancer. The presence of the APC-/- genotype has been related to the appearance of early adenomas (type 3) (*Armaghany et al., 2012*). Cells that have not inactivated APC but are characterized by KRAS activation (types 4 and 5) have been linked to aberrant crypt foci (*Pretlow and Pretlow, 2005*; *Jass, 2006*). The combination of both types of mutations (type 6) is thought to correlate with the growth of advanced adenomas (*Armaghany et al., 2012*). As in the previous study (*Paterson et al., 2020*), we assume flexibility regarding the order with which the different mutations can occur. Hence, it is assumed that the initial mutation can occur either in APC or in KRAS. It is, however, controversial whether adenoma formation can indeed be initiated by a mutation in KRAS. Some studies indicate that an initial mutation in KRAS leads to the formation of non-dysplastic polyps, which could represent an evolutionary dead end for neoplasias (*Jen et al., 1994*; *Chan et al., 2003*). On the other hand, it has been suggested that an initial KRAS mutation might be able to drive the initiation of colorectal carcinogenesis (*Jass, 2006*; *Pretlow and Pretlow, 2005*), based on mutation frequencies in aberrant crypt foci and adenomas. The assumed flexibility in the evolutionary pathway of the model accommodates these conflicting notions.

While the model assumptions about the pathways to adenoma formation are clearly defined in our model, it is important to point out that there are uncertainties in those assumptions, and that there is heterogeneity in the types of mutations that can lead to colorectal carcinogenesis. For example, it has been reported that among non-hypermutated colorectal tumors, KRAS was mutated in only about 43% of patient samples (*Network, 2012*), indicating the importance of a variety of evolutionary pathways. Our model, however, does not depend on the identity of particular mutations, but assumes the occurrence of mutation types; these are the inactivation of a tumor suppressor gene (which is a loss-of-function mutation, e.g. APC-/-), and a gain-of-function mutation, which can be in KRAS or an alternative gene. Our model predictions hold as long as the evolutionary pathway to advanced adenomas involves these two types of mutational events, regardless of their identity. We note that our model does not apply to potential cases of advanced adenomas that might develop via pathways characterized by a different number or different types of initiating events.

We model the population dynamics of the colon by using a colonic crypt as a basic unit, which is similar in concept to recently published work (*Paterson et al., 2020*). Our model is related to many previous theoretical investigations of the cell population dynamics of crypts (*Komarova et al., 2002*; *Komarova et al., 2003*; *Nowak et al., 2002*; *Shahriyari and Komarova, 2013*; *Shahriyari et al., 2016*), where stem cells (SC) were assumed to acquire random mutations in a constant-population turnover (birth and death) process, and selection happened at the level of individual stem cells. Once it was discovered that there were very few stem cells per crypt (*Nicholson et al., 2018*; *Humphries and Wright, 2008*), it became clear that the evolutionary dynamics can be conveniently described at the level of crypts, because crypts are likely to be homogeneous with respect to the driver mutations. The rate at which a crypt changes its mutational status from $i$ to $j$, denoted by $R_{ij}$, depends on the population size (the number of stem cells per crypt), the mutation rate, and the relative fitness of the invading cell type compared to the resident cell type (*Komarova et al., 2003*; *Nowak et al., 2002*). The latter can be calculated from the cell displacement data reported in the literature. Cell types APC+/-, APC-/-, and KRAS + all have a selective advantage compared to the wild type, which we assume results in an increase of the SC division rate (see Section 2 of Appendix 1 for details).

Our model keeps track of crypts of different types (denoted as $n_i$ for each type $i$). Modified crypts of types APC-/- and KRAS + have been reported to undergo crypt fission; in other words, while the total population of a single crypt remains constant (even though it is populated by SCs that are fitter than the wild-type SCs), the crypt can undergo a doubling, thus increasing the total number of such modified crypts. The fission rates of different crypt types have been reported in the literature (*Paterson et al., 2020*; *Nicholson et al., 2018*; *Humphries et al., 2013*; *Baker et al., 2014*) and are denoted by $\gamma_i$; we further denote by $\delta$ the death rate of crypts, see *Birtwell et al., 2020* for the role of crypt turn-over. We model these dynamics by using the following system of ordinary differential equations:

$$\dot{n}_1 = -(R_{12} + R_{14})n_1,$$

$$\dot{n}_2 = R_{23}n_1 - (R_{23} + R_{25})n_2,$$

$$\dot{n}_3 = R_{23}n_2 - R_{36}n_3 + \gamma 3 n_3 \left(1 - \frac{n_3+n_4+n_5}{K_A}\right) - \delta n_3$$

$$\dot{n}_4 = R_{14}n_1 - R_{45}n_4 + \gamma 4 n_4 \left(1 - \frac{n_3+n_4+n_5}{K_R}\right) - \delta n_4$$

$$\dot{n}_5 = R_{25}n_2 - R_{45}n_4 - R_{56}n_5 + \gamma 5 n_5 \left(1 - \frac{n_3+n_4+n_5}{K_R}\right) - \delta n_5,$$

where on the left hand side we have the rate of change for the population of crypts of each type, and the terms with the carrying capacity ($K_A$ for KRAS$^-$ crypts and $K_R$ for KRAS$^+$ crypts) represent competition among modified crypts that undergo crypt fission; in reference (*Paterson et al., 2020*) no crypt competition was included, such that $K_R = K_A = \infty$ in that model. The initial conditions for the system above are given by $n_1(0) = N_{crypt}$, $n_i(0) = 0$, $1 < i \leq 5$, that is, initially all $N_{crypt}$ crypts are wild type. Parameter values are presented in *Appendix 1—table 2*.

The probability to have produced a single crypt of type 6 (the APC$^{-/-}$KRAS$^+$ phenotype) by time $t$ is denoted by $P(t)$ and is approximated by the following equation *Chou and Wang, 2015*,

$$\dot{P} = (R_{56}n_5 + R_{36}n_3)(1 - P), \quad P(0) = 0$$

We further assume that crypts of type 6 engage in a fission-death dynamics (with the corresponding rates $\gamma_6$ and $\delta_6$). At the time of detection, an advanced adenoma is characterized by a certain size, $N$. If $\Delta T$ denotes the expected time for the crypt population, $n_6$, to grow to the size of detection, then the value $P(t-\Delta T)$ calculated above approximates the mathematical expression for the age-incidence curve for advanced adenoma. These approximations were checked against stochastic (Gillespie) simulations recording the incidence of size $N$ colonies of type 6 crypts, yielding excellent agreement (see Section 5 of Appendix 1 for details).

## Fitting the adenoma incidence curve

Until recently, most of the parameters associated with cellular dynamics in colonic crypts were unknown, but presently many of the rates have been estimated with a high degree of confidence (*Paterson et al., 2020*), which makes it possible to parameterize the model and use it to answer questions about the process of crypt transformation and the dynamics of cancer initiation. Using the published data on the mutation rates, the total number of crypts, the number of SCs per crypt, and the relative fitness of different cell types (see *Appendix 1—table 2*), we first attempted to fit the model in the absence of crypt competition ($K_R = K_A = \infty$), by varying the SC division rate within the physiological range and finding the best fitting value for crypt fission rates. The best fitting parameter combinations always corresponded to zero crypt fission rates. Non-zero crypt fission rates resulted in a much steeper rise in the adenoma incidence compared to that reported in *Brenner et al., 2014*. A similar result was obtained when we used different values for fitness differences (the exhaustive parameter search and a model selection procedure are described in Section 3 of Appendix 1). Finally, using the reported crypt fission rates (*Appendix 1—table 2*) we were not able to find a SC division rate within the biologically applicable range that would give the correct shape of the advanced adenoma incidence curve. The conclusion is that an unlimited exponential expansion of crypts by fission gives an unrealistically steep rise in incidence. This problem did not occur in reference (*Paterson et al., 2020*) because fitting of the age-specific incidence of CRC was not attempted, and instead, only the total life-time risk of CRC was compared to the model prediction.

Including crypt competition in the model has resolved this issue. We fixed the carrying capacity of type 3–5 crypts (parameters $K_R$ and $K_A$) to values much smaller than the initial number of healthy crypts, $N_{crypt}$, to ensure the presence of significant competition among the partially transformed crypts. Using this model, we were able to fit the data for a wide range of the SC division rates, with the non-zero best-fitting crypt fission rates that have the correct order of magnitude. Additionally, fixing the crypt fission rates to their reported values, we were able to find very well-fitting incidence curves for a wide range of SC division rates ($r_1$), with the carrying capacity parameters ranging between about 100 and about 5000.

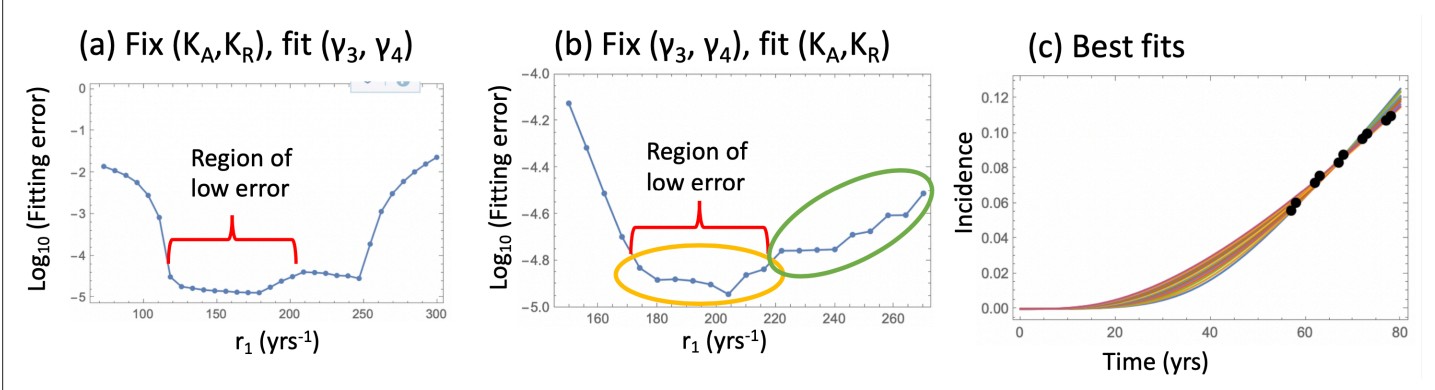

**Figure 2.** Fitting the nonlinear model to late adenoma incidence data. (**a**) The fitting error as a function of $r_1$, for the best fitting pairs ($\gamma_3$ and $\gamma_4$) with $K_A = K_R = 1000$; see *Figure 2—source data 1*. (**b**) The fitting error as a function of $r_1$, for the best fitting values of ($K_A, K_R$). Two groups of fits are marked with yellow and green ovals for the analysis of pathways, *Figure 3*. See *Figure 2—source data 2*. (**c**) The best fitting curves corresponding to increasing SC division rates, $r_1$, are plotted together with the epidemiological data (the values of $r_1$ are taken from the Region of low error, panel (**b**)). In all panels, $\delta = \delta_6 = 0.05$ yrs$^{-1}$, $\gamma_6 = 1.01$ yrs$^{-1}$, expansion of $n_6$ to $N = 10^2$ crypts, and the rest of the parameters are as in *Appendix 1—table 2*. The code for panels (**a–c**) is provided, see *Source code 1*.

The online version of this article includes the following source data for figure 2:

**Source data 1.** Data for *Figure 2a*.

**Source data 2.** Data for *Figure 2b*.

For the model that includes crypt competition, it was possible to find nearly equally good fits for a range of biologically plausible parameter values, see *Figure 2*. The amount of data in the advanced adenoma incidence curve does not allow finding unique values for all the parameters, but instead it allows using many of the parameters fixed to their experimentally obtained values, and just fine-tuning the small number of remaining parameters whose value is unknown (such as $K_A$ and $K_R$) or only its range is known (such as the SC division rate). When using the parameterized model to study the role of aspirin, instead of selecting the best fitting parameter set, we included a number of parameter sets from the best fitting parameter ranges, to see how this variability influences the result.

### Pathways to adenoma

Next, we asked what is the most likely pathway that leads to the creation of the type 6 (advanced adenoma). It is possible that crypts of type 6 could be created by a KRAS mutation in a crypt of type 3 (we called this 'APC path'), or by an APC mutation in a crypt of type 5 ('KRAS path'), see panel (a) of *Figure 3*. We found, consistent with (*Paterson et al., 2020*), that the likelihood of each of these two pathways is determined by the crypt fission rates, and not by mutation rates or crypt conversion rates (*Figure 3(d)*); in addition, it is sensitive of the carrying capacity parameters, $K_A$ and $K_R$. This is demonstrated in *Figure 3* by examining the best fitting parameter sets of *Figure 2(b)*. They naturally fall into two groups (circled in yellow and green): for the former group, the best fitting carrying capacity satisfy $K_A > K_R$, and for the latter group this inequality is reversed (see *Figure 3(b)*). We consider the former group biologically relevant, not only because it yields a smaller fitting error, but also because it corresponds to the type (APC$^{-/-}$, KRAS$^-$) crypts having a larger carrying capacity, which is consistent with this type being a more advanced stage.

The model allows for the calculation of the probabilities to develop an advanced adenoma through the APC-/- and KRAS pathways, functions $P_{APC}(t)$ and $P_{KRAS}(t)$. Panel (c) of *Figure 3* plots these quantities as functions of age (t) for the two groups of parameters. We observe that for the biologically relevant group where the carrying capacity associated with type 3 (APC$^{-/-}$, KRAS$^-$) is larger, the pathway through the inactivation of the APC gene is predominant. This is consistent with the conclusions of reference (*Paterson et al., 2020*). We also generated probability distributions of the numbers of type 3 and type 5 crypts at the time when the first type 6 crypt is generated (*Appendix 1—figures 12 and 13*). We observe that in the model, the number of type 3 (APC$^{-/-}$, KRAS$^-$) crypts is in the hundreds while type 5 (APC$^{-/+}$, KRAS$^+$) crypts are relatively rare. This might further argue against the importance of KRAS as an initiating event in disease evolution.

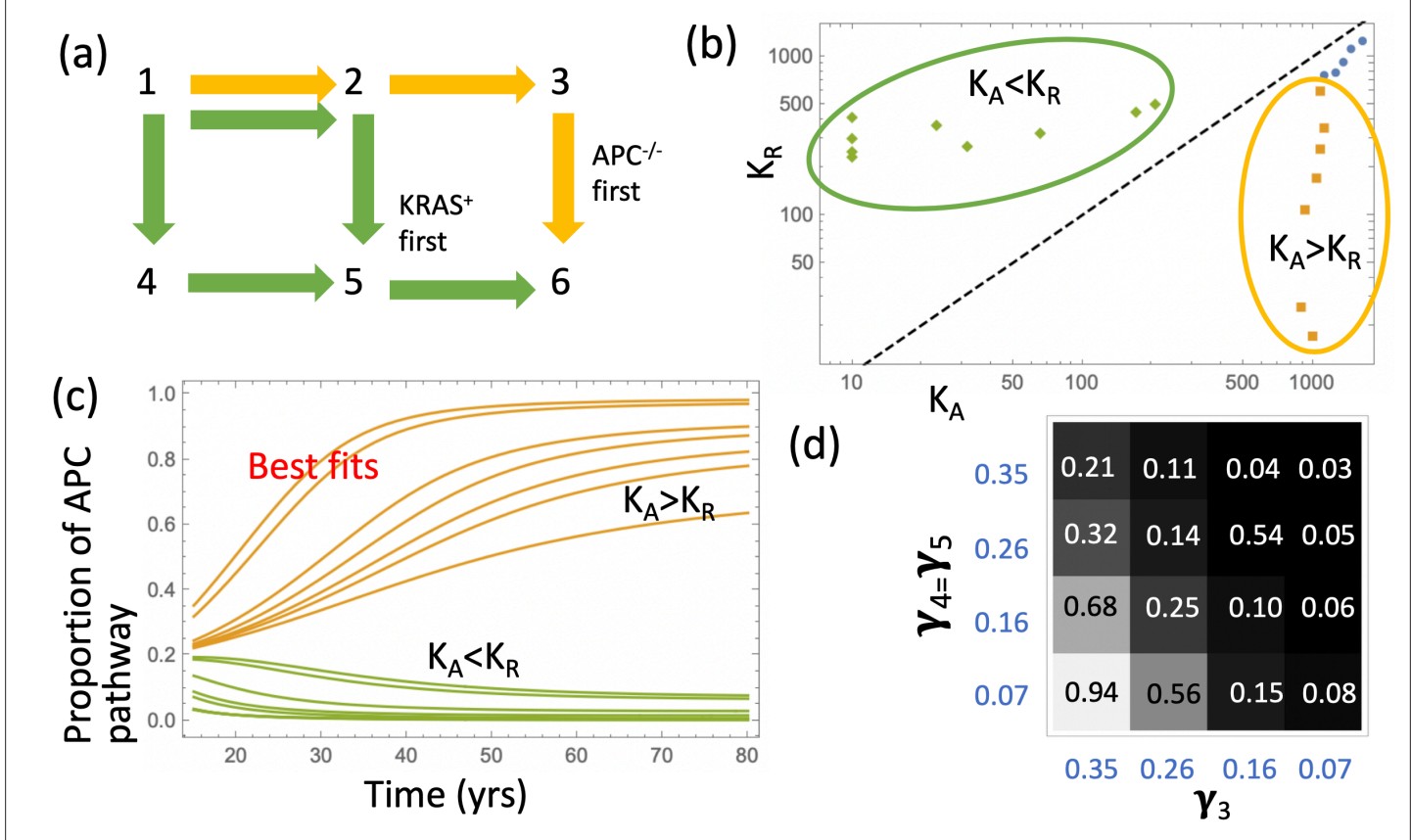

**Figure 3.** Pathways to adenoma. (**a**) A schematic representation of the two pathways. (**b**) For the two groups of fits in **Figure 2(b)**, the pairs $(K_A, K_R)$ are shown. The two groups are characterized by $K_A > K_R$ and $K_A < K_R$, respectively. See **Figure 3—source data 1**. (**c**) The probabilities $P_{APC}$ and $P_{KRAS}$ are plotted as functions of time for these two groups of fits. (**d**) The proportion of the APC-path is shown as a function of the crypt fission rates, $\gamma_3$ and $\gamma_4 = \gamma_5$. Expansion to $10^2$ type-6 crypts is assumed (see **Appendix 1—figure 10** for expansion to $10^5$ type-6 crypts); the rest of the parameters are as in **Appendix 1—table 2**. The code for panels (**b–c**) is provided, see **Source code 1**.

The online version of this article includes the following source data for figure 3:

**Source data 1.** Date for **Figure 3b**.

## The effect of aspirin

We asked, given that a variety of parameter values could lead to the same incidence curve, can we still say anything about the possible role of aspirin in cancer prevention/delay? To model the effect of aspirin on the relevant kinetic parameters, we used a variety of sources. One type of data was obtained by us in our earlier studies, where the effect of aspirin was quantified by measuring cells' kinetic parameters with and without aspirin treatment, in vitro and in xenografts (**Shimura et al., 2020**; **Zumwalt et al., 2017**). In other work, it has also been demonstrated that a related non-steroidal anti-inflammatory drug, sulindac, inhibited the fission of *APC*-deficient crypts and thus reduced adenoma numbers in mice.

It is, however, unclear which exact cell populations aspirin might affect in vivo. Therefore, we implemented the effect of aspirin in the epidemiological model by testing different sets of assumptions: (a) aspirin affects the fitness of cells within crypts (*intra-crypt dynamics*), and it may or may not affect crypt turnover dynamics through crypt fission and death rates (*inter-crypt dynamics*); (b) the fitness of type 6 cells is reduced, and the fitness of type 2–5 cells may or may not be reduced as well. In other words, only the most transformed cell type (that is, the most modified cell type that combines both the APC-/- mutation and the KRAS + mutation) is affected by aspirin, or all mutated cells, that is types 2—6, are affected. Different combinations of these assumptions have been explored, as summarized in **Figure 4(a)**.

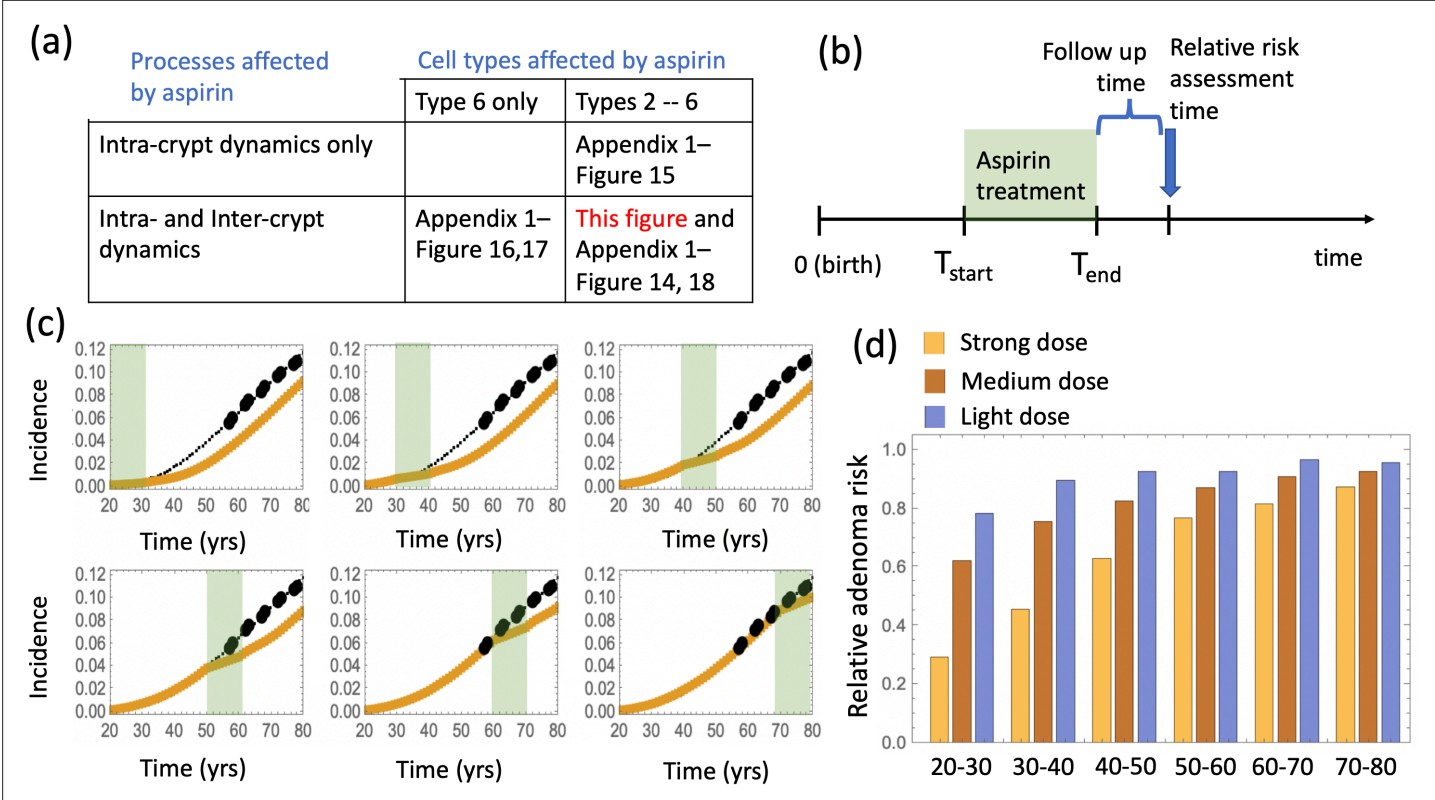

**Figure 4.** The effect of aspirin on the advanced adenoma incidence curve. (**a**) Modeling assumptions and references to figures that present the results. (**b**) A schematic showing the timing of the model with aspirin treatment. (**c**) Predicted advanced adenoma incidence in the absence of aspirin treatment (thin black lines are the fitted curves and black dots are incidence data); and under aspirin treatment where the drug affects both inter- and intra-crypt dynamics of types 2 to –6 (yellow line). Each panel corresponds to aspirin treatment administered during one decade (20–30 years, 30–40 years, etc). The treatment period is shaded light green. See *Figure 4—source data 1*. (**d**) Relative incidence of advanced adenoma, where strong (yellow, the same as in panel (**c**)), medium (brown) and light (blue) aspirin treatment dose was used (*Table 1*). Treatment is applied for different decades (as marked under the bars), and the relative risk is evaluated at the end of the treatment decade (zero follow-up time). See *Figure 4—source data 2*. Parameter set #2 (*Appendix 1—tables 1–3*) was used. $2.5 \times 10^5$ and $5 \times 10^5$ independent simulations were used for each condition in panels (**c**) and (**d**) respectively. The code for panels (**c–d**) is provided, see *Source code 2* (Mathematica). See also *Source code 3* (Fortran), a stochastic simulation that produces simulated late adenoma incidence.

The online version of this article includes the following source data for figure 4:

**Source data 1.** Data for *Figure 4c*.

**Source data 2.** Data for *Figure 4d*.

The effect of aspirin on cellular kinetics is modeled by using results of our xenograft experiments (*Shimura et al., 2020*), where we documented a dose-dependent reduction in the cell division rate (fold difference $F_r$ <1) and an increase in the cell death rate (fold difference $F_d$ >1), see *Table 1*. The relationship between the experimentally used aspirin doses in mice (*Shimura et al., 2020*) and the number of aspirin pills per weeks in humans (*Nair and Jacob, 2016*) is also given in *Table 1*. The strongest dose we used in study (*Shimura et al., 2020*) roughly translates to 1–2 standard aspirin pills

**Table 1.** Aspirin doses in xenograft experiments, the equivalent human dose and the resulting changes in kinetic rates.

| Dose in xeno-grafts (mg/kg) | Equivalent dose in humans (325 cm pills per week) | Fold difference in division rate, $F_r$ | Fold difference Un death rate, $F_d$ | Fitness factor |
|---|---|---|---|---|
| 15 | 1.8 | 0.9 | 1.5 | 0.86 |
| 50 | 6.1 | 0.75 | 1.75 | 0.70 |
| 100 | 12.2 | 0.5 | 2.0 | 0.45 |

a day for humans, which is the second strongest dose considered in *Chan et al., 2004* (6–14 pills per week), see Section 6 of Appendix 1. When aspirin is applied in our model, we assumed the following:

## Effect on the intra-crypt dynamics

For the purposes of our model, it is the combined effect of aspirin on cell division and death rates that changes the cells' relative fitness and decreases the probability of crypt conversion. To translate this information into the fold decrease in SC fitness, we note that, while the fold-reduction in division rate could be directly implemented, an increase in death rate is less straightforward. This is because in contrast to cell lines, with SCs, cell removal can occur through a combination of apoptosis and loss through differentiation, which might be the dominant component in the colorectal tissue. Therefore, if the rate of SC apoptosis is increased, say, two-fold in the presence of aspirin, this does not translate to a two-fold reduction in SC fitness. In the extreme scenario of zero SC death in the absence of aspirin, a two-fold increase in this parameter will not lead to a change in SC fitness. To calculate the fitness factor, we assumed that the removal rate of SCs, $d$, is comprised of 90% differentiation and 10% apoptosis, and that it is the latter that is affected by aspirin. If in the absence of aspirin, cellular fitness is given by the ratio r/d, then in the presence of aspirin this changes to $r/d \times F_r/(0.9 + 0.1F_d)$, which gives the fitness factor in *Table 1*. This factor enters into the crypt conversion rate, see Section 2 of Appendix 1. In particular, if only type 6 is affected, then rates $R_{36}$ and $R_{56}$ will experience a reduction. If types 2 to –6 are affected, then all conversion rates will be reduced.

## Effect of inter-crypt dynamicsf

In addition to affecting cellular fitness within the crypts, it is also logical to assume that aspirin reduces crypt fission rates and increases crypt death rates (the bottom row of the table in *Figure 4(a)*). This is supported by data (*Fischer et al., 2014*), and the rationale behind this assumption is that crypt fission is ultimately connected with divisions of individual cells, and crypt death is associated with cell death. Therefore, we assume that under aspirin treatment, $\gamma_i \to F_r \to \gamma_i$ and $\delta \to F_d\delta$ (that is, the fold-differences apply to the crypt fission and death rates). Again, this could affect the most modified crypts only (type 6), thus reducing the rate $\gamma_6$ and increasing the death rate $\delta_6$; alternatively, this could affect to all type 2 to –6 crypts, thus reducing all the crypt fission rates and increasing all the crypt death rates.

The delaying effect of aspirin was studied by using stochastic (Gillespie) simulations, where the models were run according to the schematic in *Figure 4(b)*. In particular, model parameters were switched from their (best fitted, aspirin-free) values to their modified values for the duration of treatment from $T_{start}$ to $T_{end}$. Simulations were stopped when the colony of type-6 crypts grew to its detection size (or when T = 80 was reached). Many simulation runs were aggregated to derive the age-incidence curve for advanced adenoma. *Figure 4(c)* presents typical simulation results for the models where aspirin affects both intra-crypt (conversion) and inter-crypt (crypt fission and death) dynamics for all the mutated cell types. The thin black lines represent the incidence curve in the absence of treatment, as was obtained by fitting the advanced adenoma data from *Brenner et al., 2014* (black dots). The yellow curves represent the predicted age-incidence for individuals who were undergoing aspirin treatment during the time-window (of 10 years) in different decades of their lives (see the green shading in each panel representing treatment). Panel (d) of *Figure 4* shows the relative risk of advanced adenoma for patients that received aspirin treatment during different decades of their lives. This is calculated at the time-point that is referred as 'Relative risk assessment time' in panel (b), and corresponds to a zero follow up time in this case. The yellow bars correspond to the prediction for individuals treated with a relatively strong dose of aspirin (roughly 6–14 pills a week) for 10 years. We have further performed simulations to obtain model predictions pertaining to lower aspirin doses (*Table 1*), which we referred to as medium (about 6 pills a week, brown bars in *Figure 4(d)*), and light (less than two pills a week, blue bars in *Figure 4(d)*).

We can compare the predicted relative adenoma risk with the data reported in the literature, see e.g. (*Chan et al., 2004*; *Cole et al., 2009*; *Cao et al., 2016*; *Drew et al., 2016*). In particular, the dose-dependence of colorectal adenoma was studied (*Chan et al., 2004*), and it was shown that the relative risk for adenoma was 0.80 for women who used 0.5–1.5 standard (325 mg) tablets per week, 0.74 for those who used 2–5 tablets per week, 0.72 for those who used 6–14 tablets per week, and 0.49 for those who used more than 14 tablets per week. Comparing this with the relative advanced

adenoma risk plot in *Figure 4(d)*, we can see that the model predictions are consistent with the observed bounds: for the aspirin dose that corresponds to 6–14 tablets per week (referred to as high dose, *Figure 4(d)*), the predicted relative risk ranges between about 0.35 and 0.80, depending on the age when aspirin was administered. Consistently with (*Chan et al., 2004*), we found that the effect of aspirin decreases with smaller doses, resulting in the relative risk for intermediate doses varying roughly in the 0.6–0.9 range (brown bars in *Figure 4(d)*), while for the light dose it varied in the 0.8–0.9 range (blue bars in *Figure 4(d)*).

Note that in our simulations for different scenarios, we observed remarkable quantitative consistency of results over a very wide range of parameters where uncertainties exist. For example, in *Appendix 1—figure 14*, we compared two markedly different assumptions on the numbers of type-6 crypts that constitute advanced adenoma at detection (*Lang et al., 2020*; *Sun et al., 2014*; *Kim et al., 2007*; *Tsai and Strum, 2011*; *Jones et al., 2008*; *Kang and Shibata, 2013*; *Dewanji et al., 2011*), with the results remaining very similar.

While in *Figure 4(c and d)* we assume that aspirin treatment lasts 10 years, in *Appendix 1—figure 18* we investigated the effect of a shorter duration of aspirin treatment for the high dose regime and found the relative risk closer to 0.80. One conclusion that follows from this and other simulations (see Appendix 1, section 6) is that the extent of the aspirin-induced reduction in the adenoma risk, resulting from the mechanisms studied here, is consistent with the reported risk reduction at least for a subset of the parameter combinations. Therefore, this mechanism cannot be rejected based on the predicted advanced adenoma incidence reduction.

Next, we examined how changing model assumptions about the effect of aspirin alter these results. While the simulations of *Figure 4* assume that both inter- and intra-crypt dynamics are affected by aspirin, *Appendix 1—figure 15* only includes aspirin's effect on intra-crypt dynamics. Under this assumption, even the strongest aspirin dose did not result in the magnitude of the effect reported in *Chan et al., 2004*: the reduction in advanced adenoma risk was within a few percent only. This suggests that including aspirin's effect on inter-crypt (fission/death) dynamics is essential to explain the data, which makes intuitive sense because crypt dynamics are thought to be drivers of disease development.

We also investigated the consequence of the assumption that aspirin only affects the most transformed (type 6) cells (*Appendix 1—figure 16*), and found that while the effect is reduced compared to the full model of *Figure 4*, one still observes a significant decrease in advanced adenoma risk. Interestingly, if assessment time follows treatment immediately (zero follow-up time) then there is almost no difference between the prediction of the model where only type 6 is affected compared to that where crypts 2—6 are affected (*Appendix 1—figure 17*). Increasing the follow-up time, however, reveals an increase of the difference between the two model predictions. For example, with a 15 years follow-up time, the model where all types 2—6 are affected shows the relative risk of about 0.6, while the model with only type 6 affected yields a relative risk of about 0.8. The reason for this is the lag-phase that exists between the generation of the first type-6 crypt and its growth to detection (which in our simulations takes between about 5 and 11 years).

There are several further patterns that emerge. We observe that risk reduction clearly depends on the age of the patients when aspirin was administered. As we see in *Figure 4(d)*, the relative risk can be as low as about 0.4 for patients that started treatment at age 20 and assessed at age 30, compared to a more modest reduction to relative risk of 0.8 for patients receiving treatment later in life (see additional discussion below). Further, we will mention that aspirin-induced risk reduction, as predicted by this model, does not disappear even decades after aspirin treatment stopped (*Figure 4(b)*).

Finally, we comment on another aspect of our model that is different from several other models used in the field (including *Luebeck and Moolgavkar, 2002*; *Meza et al., 2008*; *Paterson et al., 2020*). When predicting the age-incidence curve that results from the microscopic dynamics of selection and mutations, we explicitly included the growth of the most modified crypt type to its detectable size. While stopping the simulations once the first type-6 crypt is produced leads to qualitatively similar results, the inclusion of a relatively slow growth of the adenoma significantly changed numerical values of the fitted parameters. It appears that including this stage in the simulations helps improve the quantitative contribution (rather than a proof-of-principle) of this style of mathematical modeling.

## Discussion

We used mathematical modeling approaches to test the hypothesis that the changes in tumor cell kinetics observed during aspirin treatment in vitro and in vivo can translate into a protective effect on a population level that is consistent with epidemiological observations for late adenoma. This was done by first constructing a mathematical model of in vivo carcinogenesis describing evolutionary events leading up to the late adenoma stage. This model was then used to calculate expected population incidence as a function of age. Many of the model parameters have recently been estimated experimentally, which provides a solid basis for this modeling effort. Remaining parameters were estimated by fitting the incidence prediction to epidemiological data on late adenoma detection. A linear model that did not include inter-crypt competition was rejected because its best fits corresponded to zero crypt fission rates, and the more (statistically) powerful model was adopted instead, where individual mutated crypts experienced both fission and nonlinear competition dynamics. This parameterized model was used as a basis to explore how changes in the kinetics / fitness of cells, brought about by aspirin, can modify the predicted incidence of late adenomas.

Our previous in vitro and in vivo work (*Shimura et al., 2020*; *Zumwalt et al., 2017*) indicated that aspirin reduces the rate of colorectal tumor cell division and increases the rate of tumor cell death in a dose-dependent way, by up to twofold. In the current modeling study, three different experimental aspirin doses (converted to human aspirin intake) were explored, for which we previously measured their effect on the kinetic parameters. The mathematical analysis demonstrated that parameter changes of a magnitude that lies within our experimentally observed range can lead to significant reductions in predicted late adenoma incidence, which are consistent with the epidemiologically observed reductions (ranging between 10–50%, *Chan et al., 2004*). The model identified dose, treatment duration, and the age at which treatment was started as important determinants of protection in this context. We can conclude that the aspirin-induced changes in cellular fitness that we observed experimentally can in principle explain a significant portion of the protective effect observed on the population level.

This does of course not preclude alternative mechanisms that can further contribute to the protective effect. It is very likely that a reduction in the level of inflammation within the microenvironment of the cells can reduce the incidence of colorectal cancer, because inflammation has been identified as a driver of this disease (*Itzkowitz and Yio, 2004*). Moreover, other microenvironmental factors, such as the composition of the colorectal microbiome, have been shown to influence the ability of aspirin to reduce tumor growth (*Prizment et al., 2020*; *Brennan et al., 2021*; *Zhao et al., 2020*). This is therefore also likely to play a role in explaining the epidemiological data. Quantification of these further complexities in future work will allow us to introduce these additional aspects into the modeling framework, which would result in a refinement of predictions.

As with most mathematical modeling studies, there are uncertainties in assumptions that need to be kept in mind. Our experiments (*Shimura et al., 2020*; *Zumwalt et al., 2017*) were performed with tumor cell lines, both in vitro and in mouse xenografts. While the xenografts capture a higher degree of biological complexity than in vitro experiments, cellular processes in the human colon are even more complex. Colorectal tissues and tumors are characterized by stronger cell hierarchies than our experimental system, including stem, transit amplifying, and terminally differentiated cells. Our analysis was presented under the assumption that colorectal stem cells initiate and maintain tumor growth. While our experimental system did not specifically focus on stem cells, other studies indicate that the effect of aspirin on the kinetics of stem cells in particular is similar (*Chen et al., 2018*), thus justifying model assumptions. Another uncertainty concerns the cell type in which the tumor originates, and the exact identity of the cell compartment that maintains tumor growth. While we concentrated our model description around stem cells as the cell of origin that drives disease, the model defines this population as having the ability to self-renew thus maintaining the expansion of the tumor. Hence, this cell population in the model could also correspond to compartments downstream in the differentiation pathway, such as transit amplifying cells, given the marked plasticity within the intestinal epithelium. The model is thus in principle consistent with hypotheses that colorectal cancer might have a different cell of origin (*Huels and Sansom, 2015*). Interestingly, it has been shown that aspirin had a negative impact on colon organoids derived from non-neoplastic issues, and that aspirin particularly reduced the rapidly cycling transit amplifying cell population (*Devall et al., 2021*).

Another point of uncertainty concerns the identity of the cell populations that are affected by aspirin. To address this, we made several assumptions, and results remained robust. We first assumed that aspirin influences all mutated cell populations (type 2–6). Results, however, remained fairly similar in an alternative model, where only type 6 cells (characterized by APC-/- and KRAS + mutations) were affected by aspirin, although in this case the effect of aspirin is weaker, which is not surprising given that a smaller cell population loses fitness. More crucial was the assumption that aspirin influences not only the cell dynamics themselves, but also the crypt fission dynamics, reducing the rate at which crypts divide and/or increasing crypt death rate. While this is supported by data (*Fischer et al., 2014*), further experimental investigation into the exact mechanism by which aspirin affects inter-crypt dynamics is needed to back up our modeling assumptions.

An important component of all of this work is the underlying mathematical model of in vivo adenoma formation. The assumptions about the genetic events that occur during adenoma formation are consistent with our current understanding of adenoma evolution (*Fearon, 2011*), and a similar model that also includes evolutionary events beyond adenomas has recently been published (*Paterson et al., 2020*). While the model description in our and the previously published study (*Paterson et al., 2020*) was focused on APC-/- and KRAS + mutations as initial events, the same evolutionary dynamics would occur if the identity of mutations were different, as long as the evolutionary pathway involves the inactivation of a tumor suppressor gene and the activation of an oncogene, as specified in the model description.

An important difference between our and the previous model concerns assumptions about crypt fission dynamics. The previous study (*Paterson et al., 2020*) assumed that crypt fission can occur without density-dependent effects. Using experimentally available parameter estimates, this model could account for the life-time risk of colorectal cancer. When applying a similar model to late adenoma age incidence data, however, we could not obtain a good fit for the age-incidence curve, and the best fit was in fact obtained in the absence of any crypt fission. In the absence of inter-crypt competition (that is, with unlimited crypt fission), the predicted adenoma incidence rose too sharply with age compared to epidemiological data. When introducing density-dependence into the crypt fission process, however, late adenoma age incidence data could be readily fit, and so we used this model assumption to go forward. Indeed, it is likely that density-dependent effects play a role in crypt fission, because this process is probably influenced by signaling factors that become limiting as the number of crypts increases. It would be important to verify this assumption experimentally in future work.

Finally, it is interesting to discuss the results of the ASPREE trial (*McNeil et al., 2018*; *McNeil et al., 2021*) in the context of the work presented here. This trial investigated the effect of aspirin treatment in a cohort of older individuals, 70 years or older without cardiovascular disease, dementia, or disability. It was found that cancer incidence was not significantly changed by aspirin, but that the aspirin-treated group experienced a higher rate of cancer-induced mortality. The absence of a significant effect of aspirin on cancer incidence in this study is consistent with our model predictions. Our mathematical analysis demonstrated that the effect of aspirin treatment on cancer incidence diminished when treatment was initiated in older ages. Our modeling approach, however, cannot make predictions about cancer-induced mortality, because it describes the evolutionary process up to the stage of advanced adenoma only. Our previous work (*Wodarz et al., 2017*), however, offers an interpretation of these data. Because of their advanced age, it is likely that a certain fraction of the ASPREE participants already harbored tumors that had not been detected yet due to the absence of overt clinical symptoms. In fact, a previous history of cancer was not an exclusion criterion in the trial. As the established tumors continue to grow during aspirin treatment, they likely do so with altered kinetics (reduced division rates and increased death rates, leading to a higher turnover). This means that by the time the tumor has reached a given size (e.g. at which it becomes clinically detectable), it will have undergone more cell divisions under aspirin treatment compared to the placebo group. Hence, the tumor will on average have accumulated more mutations once this detectable tumor size is reached. This in turn means that the aspirin-treated tumor might be more virulent and less responsive to therapies, resulting in more deaths. The theoretically derived notion that upon detection, an aspirin-treated tumor is more evolved than a tumor that grows without aspirin (*Wodarz et al., 2017*) is supported by the ASPREE analysis, which found that aspirin-treated patients were more likely to have metastasized cancers and stage 4 cancers compared to the placebo group (*McNeil et al., 2018*; *McNeil et al., 2021*).

In conclusion, this modeling analysis suggests that a direct impact of aspirin on the kinetics and fitness of mutated cells can significantly reduce the incidence of colorectal adenomas, with a magnitude that is consistent with epidemiological data. This highlights the importance of investigating this effect of aspirin experimentally in more detail, especially under experimental conditions that approximate cell dynamics in the human colorectal tissue with greater accuracy.

## Materials and methods

### The adenoma incidence data

In order to study the incidence of advanced adenoma, we used the data reported in *Brenner et al., 2014* for the age-ranges 55–59, 60–64, 65–69, 70–74, and 75–79. While this study provides incidence data for nonadvanced adenoma, advanced adenoma, and colorectal cancer (CRC), we focused only on the combined incidence of advanced adenoma and cancer. This assumes that individuals that have developed CRC have most likely already developed an advanced adenoma by the age of testing, and further that nonadvanced adenoma likely refers to fewer mutational steps compared to our type 6, where both the APC gene is fully inactivated and the KRAS gene is mutated. The paper reports data separately for males and females; for our purposes we combined the two values to study the average, since the model is not sufficiently detailed to distinguish between the sexes.

### Mathematical modeling

The mathematical model describes stochastic dynamics of colonic crypts. There are six types of crypts that are included in the model, which differ by their mutational status. The number of crypts of each type is denoted by $n_i$, where $i = 1$ corresponds to the wild-type crypts, $i = 2$ to type APC$^{+/-}$, $i = 3$ to APC$^{-/-}$, $i = 4$ to KRAS$^+$, $i = 5$ to KRAS$^+$APC$^{+/-}$, and $i = 6$ to KRAS$^+$APC$^{-/-}$ (the most modified type associated with advanced adenoma). The model contains the processes of crypt conversion (whereby a mutation in a stem cell can fixate in a given crypt thus changing its mutational status), as well as crypt fission/death processes. Inter-crypt competition is included by way of nonlinear (logistic) terms. Given the initial condition (wild-type crypts only) and model parameters, the model outputs the probability to observe, by time $t$, a specified population of type-6 crypts ($n_6 = N$), which is assumed to be associated with advanced adenoma detection. This represents a numerically generated age-incidence curve. The expected behavior was described by a system of ordinary differential equations (ODEs), and the prediction was fitted to the advanced adenoma incidence reported in *Brenner et al., 2014*. While the model was parameterized by using the rates found in literature and describing the kinetics in humans, a subset of parameters are unknown (or only their ranges are known); these parameters were estimated by the fitting procedure.

Using the parameterized model that is consistent with the advanced adenoma incidence, we incorporated the effect of aspirin by adjusting the kinetic parameters of cells (division and death rates of cells, which describes the effect of aspirin on intra-crypt dynamics), as well as kinetic rates of crypts (crypt fission and death rates, which describes the effect of aspirin on inter-crypt dynamics). This was done by using experimentally measured factors. Fully stochastic (Gillespie) simulations were used to quantify the predicted advanced adenoma incidence curves for patients that used different doses and durations of aspirin treatment. For further details of the modeling, see Appendix 1.

## Acknowledgements

We thank the reviewers of this paper for very valuable comments that have influenced to the current form of the manuscript.

Support of the following grants is gratefully acknowledged: NIH 1 U01 CA187956-01 (AG, RB, NK, DW); NSF-Simons Center for Multiscale Cell Fate Research (NK, YW); NIH/NCI U54-CA217378 (NK, DW, YW).

## Additional information

### Funding

| Funder | Grant reference number | Author |
|---|---|---|
| National Cancer Institute | NIH 1 U01 CA187956-01 | C Richard Boland<br>Ajay Goel<br>Dominik Wodarz<br>Natalia L Komarova |
| National Science Foundation | NSF-Simons Center for Multiscale Cell Fate Research | Yifan Wang<br>Natalia L Komarova |
| MIDAS | AWD00000238 | Natalia L Komarova<br>Yifan Wang |

The funders had no role in study design, data collection and interpretation, or the decision to submit the work for publication.

### Author contributions

Yifan Wang, Formal analysis, Methodology, Software, Visualization, Writing – original draft, Writing – review and editing; C Richard Boland, Ajay Goel, Conceptualization, Funding acquisition, Writing – review and editing; Dominik Wodarz, Conceptualization, Funding acquisition, Methodology, Writing – original draft, Writing – review and editing; Natalia L Komarova, Conceptualization, Formal analysis, Funding acquisition, Investigation, Methodology, Project administration, Supervision, Validation, Writing – original draft, Writing – review and editing

### Author ORCIDs

Yifan Wang ⓘ http://orcid.org/0000-0003-3292-4972
C Richard Boland ⓘ http://orcid.org/0000-0002-8120-5088
Ajay Goel ⓘ http://orcid.org/0000-0003-1396-6341
Dominik Wodarz ⓘ http://orcid.org/0000-0002-8017-3707
Natalia L Komarova ⓘ http://orcid.org/0000-0003-4876-0343

### Decision letter and Author response

Decision letter https://doi.org/10.7554/eLife.71953.sa1
Author response https://doi.org/10.7554/eLife.71953.sa2

## Additional files

### Supplementary files

- Transparent reporting form
- Source code 1. Mathematica code for *Figures 2 and 3*.
- Source code 2. Mathematica code for *Figure 4*.
- Source code 3. A Fortran code for stochastic simulations (e.g. *Figure 4*).

### Data availability

Data and some relevant code are available on Dryad under DOI 10.7280/D1M11M. Code for Figures 2, 3 and 4 is uploaded in Source Code files 1-3.

The following dataset was generated:

| Author(s) | Year | Dataset title | Dataset URL | Database and Identifier |
|---|---|---|---|---|
| Komarova NL | 2022 | Data files for simulated advanced adenoma age incidence, under aspirin treatment | https://doi.org/10.7280/D1M11M | Dryad Digital Repository, 10.7280/D1M11M |

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

## Appendix 1

## Mathematical formulation of a crypt-based model of adenoma initiation

Let us enumerate the types in the way presented in Table *Appendix 1—table 1*.

**Appendix 1—table 1.** Enumeration of the different genotypes.

| Mutations in APC | Mutations in KRAS | Type number |
|---|---|---|
| 0 | 0 | 1 |
| 1 | 0 | 2 |
| 2 | 0 | 3 |
| 0 | 1 | 4 |
| 1 | 1 | 5 |
| 2 | 1 | 6 |

Then we can denote by $n_i$ with $1 \leq i \leq 6$ the number of crypts of type . Suppose $R_{ij}$ is the conversion rate from crypt type to type $j$, and $\gamma_i$ the growth rate (by crypt fission) of crypts of type . We have the following equations:

$$\dot{n}_1 = -(R_{12} + R_{14})n_1, \tag{1}$$

$$\dot{n}_2 = R_{12}n_1 - (R_{23} + R_{25})n_2 + \gamma_2 n_2, \tag{2}$$

$$\dot{n}_3 = R_{23}n_2 - R_{36}n_3 + \gamma_3 n_3, \tag{3}$$

$$\dot{n}_4 = R_{14}n_1 - R_{45}n_4 + \gamma_4 n_4, \tag{4}$$

$$\dot{n}_5 = R_{25}n_2 + R_{45}n_4 - R_{56}n_5 + \gamma_5 n_5, \tag{5}$$

with the initial conditions

$$n_1(0) = N_{crypt}, \quad n_i(0) = 0, \quad 1 \leq i \leq 5. \tag{6}$$

In the ODEs above, we have ignored the process of stochastic tunneling such that the crypts can only convert one step at a time. It is further possible to ignore the negative (outgoing) rates, which simplifies this linear system to the following:

$$\dot{n}_1 = 0, \tag{7}$$

$$\dot{n}_1 = R_{12}n_1 + \gamma_2 n_2, \tag{8}$$

$$\dot{n}_3 = R_{23}n_2 + \gamma_3 n_3, \tag{9}$$

$$\dot{n}_4 = R_{14}n_1 + \gamma_4 n_4, \tag{10}$$

$$\dot{n}_5 = R_{25}n_2 + R_{45}n_4 + \gamma_5 n_5. \tag{11}$$

The probability $P(t)$ that by time $t$ at least one crypt of type 6 has been created (using the mean-field approximation from *Cole et al., 2009*) is given by the solution of the equation

$$\dot{P} = (R_{56}n_5 + R_{36}n_3)(1 - P), \quad P(0) = 0, \tag{12}$$

see also section 5.1. The solution can be obtained exactly and is given by

$$P = 1 - exp\left\{-N_{crypt}(S_{45}R_{1\to4\to5\to6} + S_{25}R_{1\to2\to5\to6} + S_{23}R_{1\to2\to3\to6})\right\}, \tag{13}$$

where the quantities $S_{ij}$ correspond to the paths $1 \to i \to j \to 6$,

$$R_{1\to i\to j\to 6} = R_{1i}R_{ij}R_{j6}, \tag{14}$$

see *Appendix 1—figure 1*. They can be written down by using the following function:

$$g(x) = \frac{e^{xt}-1-xt}{x^2} > 0. \tag{15}$$

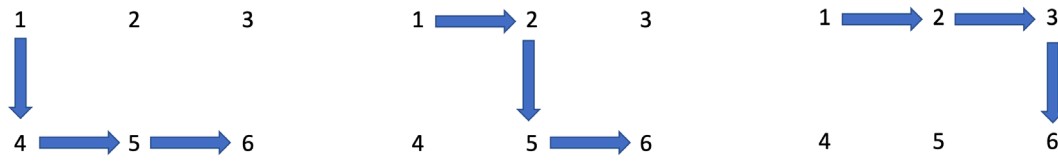

**Appendix 1—figure 1.** Three pathways to adenoma.

We have

$$g(0) = \frac{t^2}{2}, \quad g'(x) \equiv \frac{\partial g}{\partial x} = \frac{2+tx+e^{tx}(tx-2)}{x^3} > 0,$$

that is, this function increases monotonically in $x$. We have

$$S_{ij} = \frac{g(\gamma_i)-g(\gamma_j)}{\gamma_i-\gamma_j}.$$

Note that expressions $S_{ij}$ have a singularity if any of the quantities $\gamma_i$, $\gamma_j$ is zero and/or if $\gamma_i = \gamma_j$. For example, if both growth rates are zero ($\gamma_i = \gamma_j = 0$), we have

$$S_{ij} = \frac{t^3}{6}.$$

In our context it is reasonable to assume that $\gamma_2 = 0$, that is, APC+/- crypts do not divide, and that $\gamma_4 = \gamma_5$, that is, APC+/+ and APC+/- crypts with an additional KRAS mutation divide at the same rate. In the $\gamma_i = \gamma_j$ case, taking the limit as $\gamma_j \to \gamma_i$, we obtain

$$S_{45} = g'(\gamma_4), \quad S_{25} = \frac{g(\gamma_4)-t^2/2}{\gamma_4}, \quad S_{23} = \frac{g(\gamma_3)-t^2/2}{\gamma_3}. \tag{16}$$

It is convenient to view $S_{25}$ as a function of $\gamma_4$, $S_{25} = F(\gamma_4)$, where

$$F(x) = \frac{1}{x^3}\left(e^{xt} - \left(1 + xt + \frac{(xt)^2}{2}\right)\right) = \frac{1}{x^3}\sum_{i=3}^{\infty}\frac{(xt)^i}{i!},$$

then $S_{23} = F(\gamma_3)$, and $F(x)$ is an increasing function of $x$. That is, if $\gamma_4 > \gamma_3$ then $S_{25} > S_{23}$, and if $\gamma_4 < \gamma_3$ then $S_{25} < S_{23}$. We further note that $S_{45} = \tilde{F}(\gamma_4)$ with

$$\tilde{F}(x) = \frac{1}{x^3}\left(e^{xt}(tx-2) + tx + 2\right) = \frac{1}{x^3}\sum_{i=3}^{\infty}\frac{(xt)^i(i-2)}{i!} > F(x).$$

To summarize, we can see that $S_{45} > S_{25}$, and $S_{23}$ is greater (smaller) than $S_{25}$ if $\gamma_3$ is greater (smaller) than $\gamma_4$.

## Model parameters

Here we define parameters that appear in this model. We will make the following assumptions (see also Table *Appendix 1—table 2*):

- The mutation rate between types is given as follows: $u_{1\to2} = u_{4\to5} = 2u$ (inactivation of the first copy of the APC gene, that is, any of the two copies); $u_{2\to3} = u_{5\to6} = u$ (inactivation of the remaining copy of the APC gene); $u_{1\to4} = u_{2\to5} = u_{3\to6} = \mu$ (activation of the KRAS gene). The fitness of cells with APC+\−, APC−/−, and KRAS+ phenotypes relative to the wild type cells was determined using the cell replacement data from *Komarova et al., 2002*. In particular, we assumed that for any phenotype,

$$F_j \equiv \frac{r_j d_1}{d_j r_1} = \frac{Pr(j)}{1 - Pr(j)}, \quad j \in 2, 3, 4$$

where $Pr(j)$ is the probability of replacement of the wild type by type $j$ found in *Komarova et al., 2002*. Using this formula, we obtain the values given in Table *Appendix 1—table 2*. Then the fitness of other types is multiplicative: $F_5 = F_4 F_2, F_6 = F_3 F_4$.

- The death rates are assumed the same among the types, such that the division rates of cells satisfy $r_i = F_i r_1, \quad 1 < i \le 6$.
- Wild-type crypts and those with only a single copy of APC gene mutated do not proliferate ($\gamma_1 = \gamma_2 = 0$).
- Crypts with a KRAS mutation and APC+/+ and APC+/- phenotypes proliferate at the same rate, $\gamma_4 = \gamma_5$.
- All the parameters are specified in Table *Appendix 1—table 2*.

The conversion rates are given by

$$R_{ij} = r_i K u_{i\to j} \rho_{ij}, \tag{17}$$

where $r_i$ is the division of type , $u_{i\to j}$ is the mutation rate from type to type $j$, and $\rho_{ij}$ is the probability that one cell of type $j$ becomes fixated in a compartment of size $K$ with the host type . To calculate this probability, let us denote by $d_i$ the death rate of type . Then we have

$$\rho_{ij} = \frac{1 - \frac{r_i d_j}{r_j d_i}}{1 - \left(\frac{r_i d_j}{r_j d_i}\right)^K}, \tag{18}$$

where $\frac{r_i d_j}{r_j d_i}$ is the inverse of the relative fitness of type $j$ with respect to type .

**Appendix 1—table 2.** Parameters, notations, and their values.

| Parameter | Notation | Value/Range |
|---|---|---|
| Number of crypts | $N_{crypt}$ | $10^7$ |
| Number of SCs per crypt | K | 7 |
| Rate of inactivation of APC (per cell division) | u | $10^{-7}$ |
| Rate of inactivation of APC (per cell division) | μ | $10^{-9}$ |
| Division rate of WT SCs (per year) | $r_1$ | (18,365) |
| Relative fitness of APC +/- cells | $F_2 = F_{APC^{+/-}}$ | 1.6 |
| Relative fitness of APC-/- cells | $F_3 = F_{APC^{-/-}}$ | 3.76 |
| Relative fitness of KRAS+ cells | $F_4 = F_{KRAS}$ | 3.54 |
| Division rate of APC-/- crypts (per year) | $\gamma_3$ | 0.2 |
| Division rate of KRAS+ crypts (per year) | $\gamma_4$ | 0.07 |

## Fitting the linear model to advanced adenoma incidence data

The linear model of the adenoma incidence is given by *equations (13,14,15,16,17,18)*.

In order to fit this model to the data, we fix parameters $N_{crypt}, K, u, \mu$, and vary the remaining parameters. This is done in stages. We first fix the fitness parameters $F_2, F_3, F_4$ to their values in table (**Appendix 1—table 2**) and vary the remaining parameters $r_1, \gamma_3, \gamma_4$ to find the global minimum of the error between the model and the data, see **Appendix 1—figure 2a-c**. Then we take other select values for the relative fitness parameters to show that the results remain qualitatively similar (not shown). Next, we describe the results of fitting and the patterns that were observed.

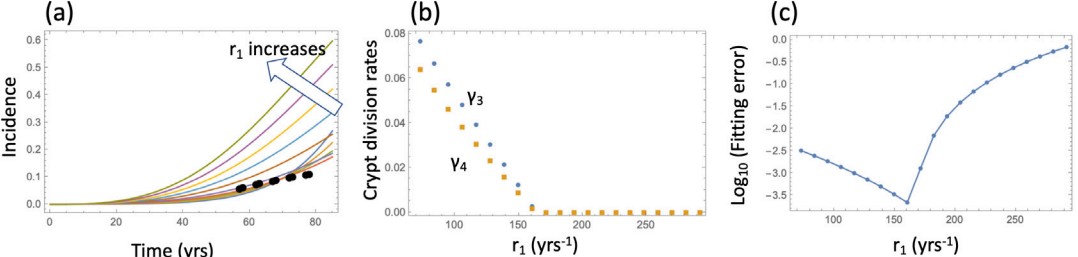

**Appendix 1—figure 2.** Model fitting to the incidence data. (**a**) The best fitting curves corresponding to increasing SC division rates, $r_1$, are plotted together with the epidemiological data. (**b**) The best fitting parameters $\gamma_3$ and $\gamma_4$ are shown for each value of $r_1$. (**c**) The fitting error as a function of $r_1$. The relative fitness values are fixed to $F_{APC+/-} = 1.6, F_{APC-/-} = 3.76, F_{KRAS} = 3.54$. The rest of the parameters are as in Table **Appendix 1—table 2**.

To find the global minimum of the error, we varied parameter $r_1$ (the division rate of the wild type SCs) between once a day and once every 20 days (which corresponds to the division rate of 365 yrs$^{-1}$ and 18 yrs$^{-1}$). For each value of $r_1$, the error was minimized in the 2-dimensional parameter space ($\gamma_3, \gamma_4$), and a unique minimum was always found. The best fits corresponding to a subset of these division rates are plotted in **Appendix 1—figure 2(a)**, with the best fitting values of $\gamma_3$ and $\gamma_4$ shown in panel (b) for each $r_1$. We can see that as $r_1$ increases, that is, SCs are assumed t divide faster, the best fitting crypt division rates decrease (that is, crypt fission proceeds at a slower rate). For $r_1$ greater than about 250 yrs$^{-1}$ (that is, divisions once every day and a half or faster), the best fitting crypt fission rates are zero. This parameter combination (division rate of about every 1.5 days and zero crypt fission) corresponds to the minimal error of fitting (panel (c)). This is evident from the shape of the best fitting incidence curves, $P(t)$, corresponding to different $r_1$ values (panel (a)). For low rates of SC division, the fission rates are relatively high, resulting in an curve $P(t)$ that has a steep rise, yielding a large fitting error and a qualitatively unrealistic shape of the incidence curve if compared with the data. This is the consequence of a pronounced exponential increase in the number of crypts, which sharply accelerates the generation of type 6 crypts. As $r_1$ increases and crypt fission rates decrease, the incidence curves become less steep, until the best fitting crypt fission rate reaches zero. At this stage, the best fitting incidence curve is achieved, because further increase in $r_1$ results in an increase in the incidence that happens too early.

So far, in order to fit the model to the adenoma data, we fixed several of the parameters to their measured values and varied the three remaining parameters ($r_1, \gamma_3, \gamma_4$) within realistic ranges to investigated the error landscape and find global minima. To obtain a more comprehensive picture of the model behavior, we have implemented a procedure where five parameters were varied: the division rate of stem cells, $r_1$, two cellular fitness parameters, $R_{APC+/-}$ and $R_{KRAS}$ (with $F_{APC+/+} = 2F_{APC+/-}$); and two crypt fission parameters, $\gamma_3$ and $\gamma_4$. The rest of the parameters were set to their values in Table **Appendix 1—table 2**.

The fitness parameters were varied between 1.1 and 3.5 to match the measured range. For each pair ($R_{APC+/-}, R_{KRAS}$), a fitting procedure identical to that of **Appendix 1—figure 2** was performed, see **Appendix 1—figure 3**. Each graph corresponds to a unique pair ($R_{APC+/-}, R_{KRAS}$), and the values of these coefficients are indicated on each panel. The horizontal axes of each panel is the stem cell division rate, $r_1$. The green curves show the $log_{10}$(fitting error), and the black (gray) dots show the best fitting values of $\gamma_3$ ($\gamma_4$), multiplied by 10 for convenience of presentation. If the best fitting crypt fission parameter was negative, then the fitting error shows corresponded to $\gamma_3 = \gamma_4 = 0$. We observe that in all the cases, the best fitting parameter combination is reached when the crypt fission rates become zero, reproducing the result of **Appendix 1—figure 2**, but for a wide parameter range.

To observe this more clearly, we also presented the error landscape for the best fitting parameter $r_1$, as a function of $\gamma_3$ and $\gamma_4$ (the horizontal and vertical axes in each panel, respectively). The quantity $log_{10}$(fitting error) is shown as a heat map, with darker colors corresponding to lower error values. We can see that the lowest error corresponds to the corner $\gamma_3 = \gamma_4 = 0$.

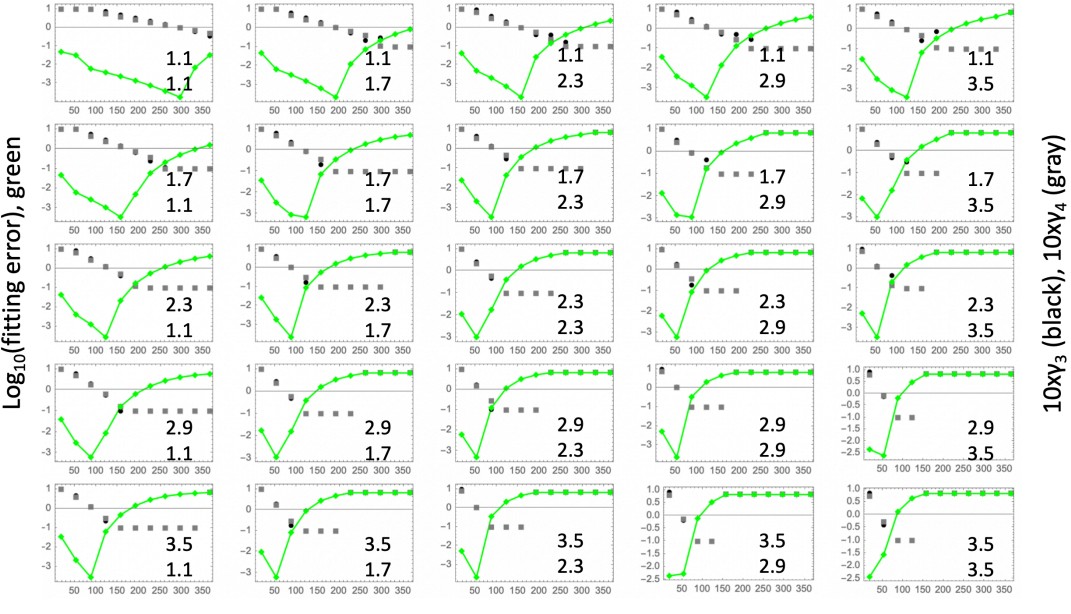

**Appendix 1—figure 3.** Fitting the linear model. Each panel shows fitting results for a particular parameter combination, $(R_{APC^{+/-}}, R_{KRAS})$; the values of these two fitness parameters are indicated, and $F_{APC^{+/+}} = 2F_{APC^{+/-}}$. The green lines show the $log_{10}$(fitting error) as a function of $r_1$. The fitted parameters $\gamma_3$ and $\gamma_4$ (multiplied by 10) are shown as black (gray) lines. The rest of the parameters are as in Table **Appendix 1—table 2**.

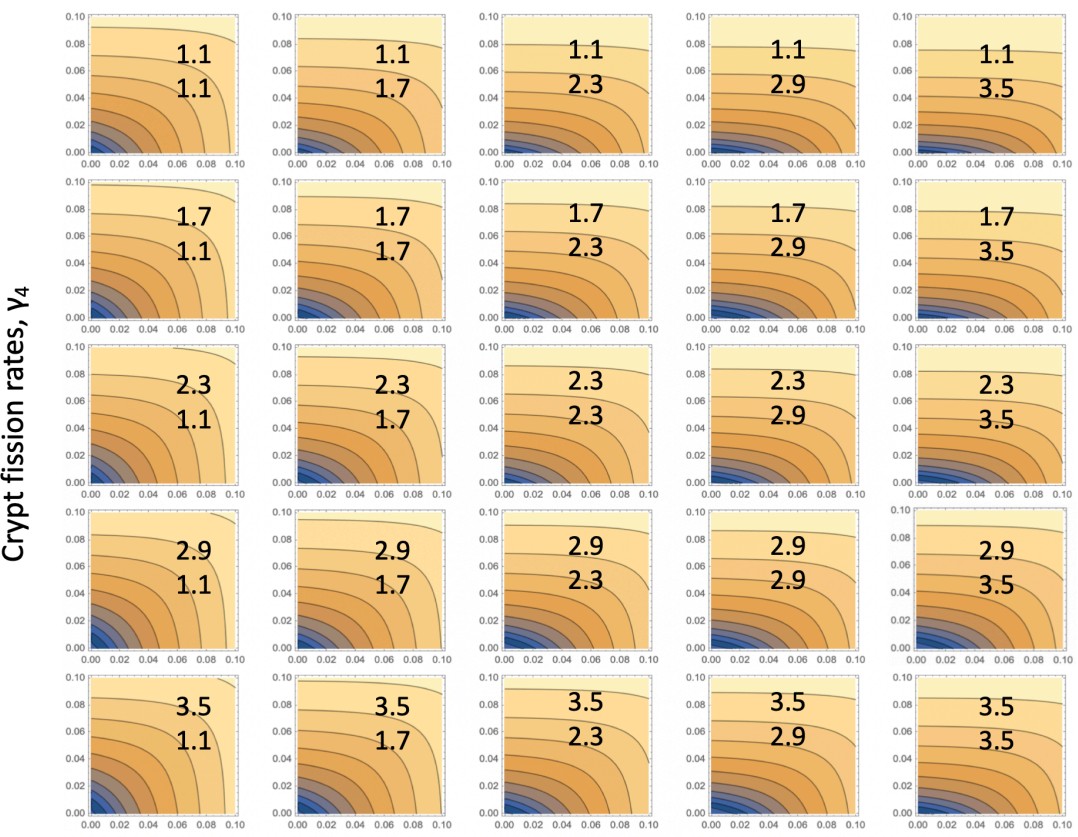

Crypt fission rates, $\gamma_4$

Crypt fission rates, $\gamma_3$

**Appendix 1—figure 4.** Fitting the linear system: for each parameter combination, $(R_{APC^{+/-}}, R_{KRAS})$, of *Appendix 1—figure 3*, the error landscape is shown that corresponds to the best fitting $r_1$ (see the minima of the green lines in *Appendix 1—figure 3*). The contour plot represents $log_{10}$(fitting error) as a function of $\gamma_3$ and $\gamma_4$ (darker colors correspond to lower values). The rest of the parameters are as in Table *Appendix 1—table 2*.

## Nonlinear model: competition among crypts

Let us assume that the types of crypts that undergo crypt fission (types 3,4, and 5) are in direct competition with each other, modeled as logistic, as opposed to straight exponential, growth. Let us denote te carrying capacity associated with the growth of the KRAS- crypt (type 3) as $K_A$, and that for the KRAS+ crypts (types 4 and 5) as $K_R$. Then the nonlinear system with crypt competition becomes

$$\dot{n}_1 = -(R_{12} + R_{14})n_1, \tag{19}$$

$$\dot{n}_2 = R_{12}n_1 - (R_{23} + R_{25})n_2, \tag{20}$$

$$\dot{n}_3 = R_{23}n_2 - R_{36}n_3 + \gamma_3 n_3 \left(1 - \frac{n_3 + n_4 + n_5}{K_A}\right) - \delta n_3, \tag{21}$$

$$\dot{n}_4 = R_{14}n_1 - R_{45}n_4 + \gamma_4 n_4 \left(1 - \frac{n_3 + n_4 + n_5}{K_R}\right) - \delta n_4, \tag{22}$$

$$\dot{n}_5 = R_{25}n_2 + R_{45}n_4 - R_{56}n_5 + \gamma_5 n_5 \left(1 - \frac{n_3 + n_4 + n_5}{K_R}\right) - \delta n_5, \tag{23}$$

with initial conditions (6) and the probability to create a crypt of type 6 given by system (12). While an analytical solution is no longer available, a procedure similar to that performed for the linear system can be performed numerically. Results can be seen in figure *Appendix 1—figure 5*. As before, all the parameters were fixed to their values in Table *Appendix 1—table 2*, except the

cell division rate, $r_1$, and the crypt fission rates, $\gamma_3$ and $\gamma_4$. Additionally, we assumed a crypt carrying capacity $K_A = K_R = K = 10^3$ and crypt death rate $\delta = 0.05$ yrs$^{-1}$. The best fitting values of $\gamma_3$ and $\gamma_4$ were found for each value of $r_1$, which was varied in the realistic range. This procedure yielded a range of low-error fits (panel (c)) that correspond to nonzero values of crypt fission rates in the realistic range (panel (b)), with the corresponding incidence curves given in the inset of panel (a).

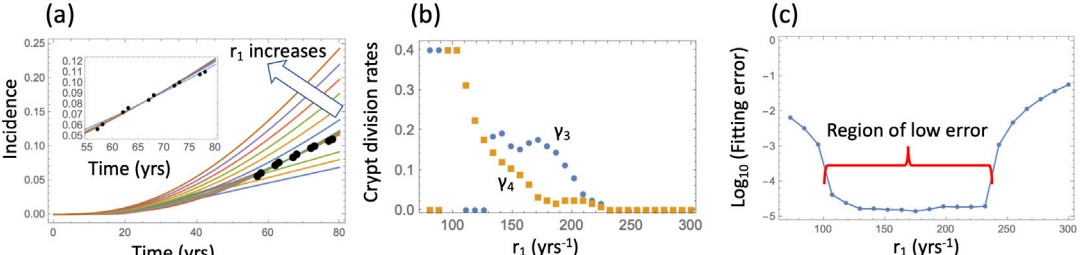

**Appendix 1—figure 5.** Nonlinear model fitting to the incidence data (fixing the crypt carrying capacity, varying crypt fission rates). (**a**) The best fitting curves corresponding to increasing SC division rates, $r_1$, are plotted together with the epidemiological data; inset: the best fitting curves corresponding to the values of $r_1$ from the Region of low error, see panel (**c**). (**b**) The best fitting parameters $\gamma_3$ and $\gamma_4$ are shown for each value of $r_1$. (**c**) The fitting error as a function of $r_1$. The relative fitness values are fixed to $F_{APC^{+/-}} = 1.6, F_{APC^{-/-}} = 3.76, F_{KRAS} = 3.54, K_A = K_R = K = 1000$. The rest of the parameters are as in Table **Appendix 1—table 2**.

Since this procedure yielded a wide range of similarly good fits under a fixed value for the crypt carrying capacity, we next performed a fitting procedure where the crypt fission rates $\gamma_3$ and $\gamma_4$ were fixed to those in Table **Appendix 1—table 2**, and the best carrying capacity, $K$, was determined for each division rate, $r_1$, by fitting to the data. Results (under the assumption that $K_A = K_r = K$) are presented in **Appendix 1—figure 6a-c**. We can see that for a range of values of the cell division rates, $r_1$, a low-error fit was found, with the carrying capacity values ranging between about $5 \times 10^2$ and $5 \times 10^3$. The best fitting values are $K = 1318, r_1 = 141.1$ (divisions approximately every 2.5 days). Changing the crypt death rate to $\delta = 0$ yields very similar results, see **Appendix 1—figure 6d-f**.

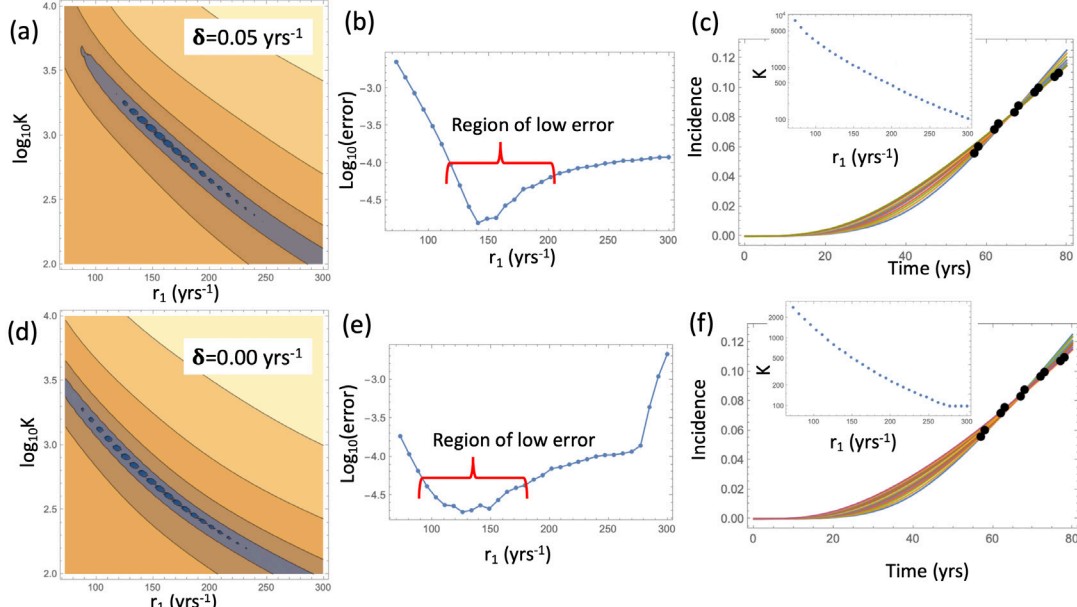

**Appendix 1—figure 6.** Nonlinear model fitting to the incidence data (fixing crypt fission rates, varying the crypt carrying capacity). Compared are the cases in the presence of a non-zero crypt death rate, $\delta = 0.05$ yrs$^{-1}$ (**a–c**), and in the absence of crypt death rate, $\delta = 0.0$ yrs$^{-1}$ (**d-f**). (**a,d**) The heatplot of the fitting error, where all parameters are fixed except $K_A = K_R = K$ and $r_1$. Dark colors correspond to lower values of the error. (**b,e**) The *Appendix 1—figure 6 continued on next page*

*Appendix 1—figure 6 continued*
fitting error as a function of $r_1$. **(c,f)** The best fitting curves corresponding to increasing SC division rates, $r_1$, are plotted together with the epidemiological data the value of $r_1$ are taken from the Region of low error. Inset: The best fitting carrying capacity $K$ is shown for each value of $r_1$. The rest of the parameter values are fixed to, $F_{APC+/-} = 1.6, F_{APC-/-} = 3.76, F_{KRAS} = 3.54, \gamma_3 = 0.2, \gamma_4 = 0.07$ and the rest as in Table *Appendix 1—table 2*.

These results suggest that biologically, the nonlinear model is a more appropriate choice because it produces the best fit for values of crypt fission rates that are within the experimentally observed range, while the linear model requires zero crypt fission rates. From the statistical point of view, the nonlinear model is a significantly more powerful model e.g. by applying the Akaike Infromation Criterion (AIC).

## Model predictions and advanced adenoma generation dynamics

### Gillespie simulations and the mean-field approximation

*Equation 12* approximates the probability $P(t)$ that by time $t$, a single type 6 crypt has been generated. In order to test its validity we have run stochastic Gillespie simulations based on system (LABEL:n1non-23), which were stopped when the first type 6 crypt was created. The resulting incidence was compared to that obtained from the deterministic system (LABEL:n1non-23, 12), see *Appendix 1—figure 7*, line (0). There, the blue symbols and the blue solid line correspond to the stochastic simulations and the deterministic approximation respectively. The deterministic approximation showed excellent agreement with Gillespie simulations.

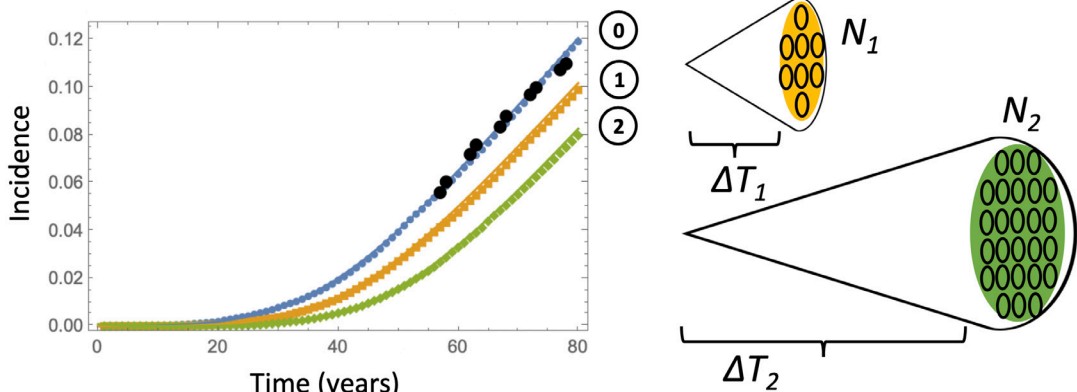

**Appendix 1—figure 7.** Comparing deterministic solutions (system (LABEL:n1non-23, 12), solid lines) with the stochastic Gillespie simulations (symbols of the same color). The three lines correspond to the different stopping conditions: (0) 1 crypt of type 6; (1) $N_1 = 10^2$ crypts of type 6, (2) $N_2 = 10^5$ crypts of type 6. The rest of the parameters are as in *Appendix 1—figure 6a-c*. 100,000 independent simulations were performed for each line, and the standard deviation bars are too small to see.

Gillespie simulations described here can be used to simulate stochastic effects associated with crypt dynamics studied here, see e.g. results presented in Section 5.3. These simulations were also used to study the effect of aspirin, Section 6. Finally, this methodology can be extended to study the clonality of abnormal crypts. The ODE model does not keep track of clonality. For example, if type 5 crypt is created multiple times (by conversion) in the system, the variable $n_5$ simply gives the total number of type-5 crypts. In a stochastic Gillespie model, however, it is possible to keep track of different clones by designating each newly generated crypt as a different "sub-type", which can then clonally expand through crypt fission. This however goes beyond the scope of the current study.

### The growth phase of advanced adenoma

So far we have stopped the simulations as soon as the first copy of a type 6 crypt was created. Alternatively, one could explicitly account for the growth phase of the APC-/- KRAS+ crypts. Estimates for the doubling time of advanced adenomas vary in the literature. For example, (*Jones et al., 2008*) suggests that the doubling time of advanced adenoma is 250 days (that is, the net rate of expansion 1.01 yr⁻¹), with a similar doubling time of about 1 year quoted in *Kang and Shibata,*

*2013* (the net rate of expansion 0.7 yr$^{-1}$). The number of crypts in an advanced adenoma can be estimated as follows. Given that the minimum advanced adenoma size is 10 mm (see *Dewanji et al., 2011* and also *Chen et al., 2018*), and assuming $10^9$ cells per cm$^3$ (*Huels and Sansom, 2015*), we obtain that each advanced adenoma contains about $10^6$ cells. This comprises about $10^2$ crypts, if we assume that type 6 crypts are $10^4$ cells each, which is somewhat larger that normal colonic crypts that measure about $2 \times 10^3$ cells (*Devall et al., 2021*). A somewhat larger estimate is given in *McNeil et al., 2018* where it is noted that a 1 mm$^3$ adenoma contains on the order of $5 \times 10^5$ cells, which translates into $5 \times 10^8$ cells in an adenoma of size 10 mm$^3$, or $5 \times 10^4$ crypts. Therefore, we have explored a range of crypt expansion to sizes $N_1 = 10^2$ and $N_2 = 10^5$ ctypts, see *Appendix 1—figure 7*. Assuming, for type 6 crypts, $\gamma_6 = 1.01$ yr$^{-1}$ and the crypt death rate $\delta_6 = 0.05$ yr$^{-1}$, we obtain that expansion from 1 type 6 crypt to $N_1$ crypts will take $\Delta T_1 = 4.79$ years, and expansion from 1 type 6 crypt to $N_2$ crypts will take $\Delta T_2 = 11.97$ years respectively.

Using these calculations, we obtain the deterministic prediction that expansion to $10^2$ crypts will shift the incidence curve by $\Delta T_1$ years, and expansion to $10^5$ crypts will shift it by $\Delta T_2$ years. Apart from this shift, the probability of crypt non-extinction has to be incorporated. If $P(t)$ is the solution of *equation (12)* under system (LABEL:n1non-23), then we can approximate the probability $P_i(t)$ that by time $t$, $N_i$ crypts of type 6 have been generated, by

$$P_i(t) = \left(1 - \frac{\delta}{\gamma_6}\right) P(t - \Delta T_i), \quad i = 1, 2. \tag{24}$$

The corresponding curves are marked as curves 1 and 2 in *Appendix 1—figure 7*, see the orange and green solid lines respectively. To check this approximation, we have run Gillespie simulations that stopped as soon as $N_1 = 10^2$ (orange symbols) and $N_2 = 10^5$ (green symbols) crypts were generated. Again, very good agreement between the stochastic simulations and our deterministic approximation was observed.

Next, we will show that similar results are obtained when the model with an explicit type 6 crypt expansion phase is fit to the advanced adenoma incidence data. Fixing a crypt carrying capacity, we found the best fitting fission rates $\gamma_3$ and $\gamma_4$ for each value of $r_1$. The procedure was similar to that used in *Appendix 1—figure 5*, except probabilities to have $N_i$ type 6 crypts, $P_i(t)$ (*equation (24)*) were used to fit the incidence data. Results are shown in *Appendix 1—figure 8*. The plot of the fitting error again shows regions of low error (panel (a)). In those regions, nonzero values of crypt fission correspond to the best fit (see panels (b) and (c), where regions of low error are shaded red). As in *Appendix 1—figure 5(b)*, the best fitting values of $\gamma_4$ are lower than those of $\gamma_3$, and both are similar to those found in the literature, see Table *Appendix 1—table 2*.

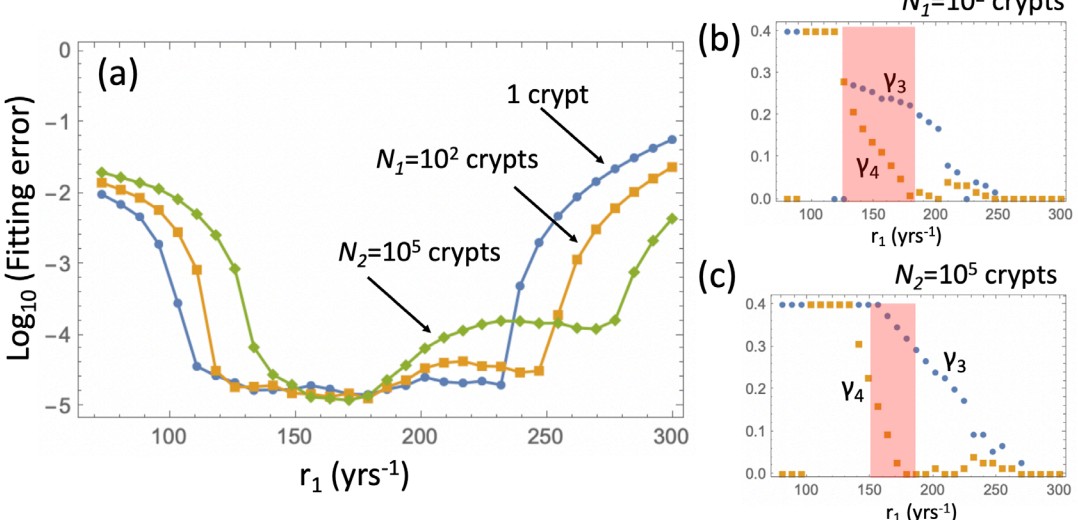

**Appendix 1—figure 8.** Fitting the adenoma incidence curve to the nonlinear model with an explicit expansion phase. Under a fixed $K_A = K_R = 1000$, for each value of $r_1$ the best fitting values of $\gamma_3$ and $\gamma_4$ were found. (**a**) The
*Appendix 1—figure 8 continued on next page*

*Appendix 1—figure 8 continued*

fitting error is shown as a function of $r_1$ for the original model (blue line, same as in *Appendix 1—figure 5(c)*); the model with expansion to $N_1$ crypts (yellow curve), and the model with expansion to $N_2$ crypts (green curve). (**b**) The best fitting values of crypt fission rates, $\gamma_3$ and $\gamma_4$, for the model with expansion to $N_1$ crypts. (**b**) The best fitting values of crypt fission rates, $\gamma_3$ and, $\gamma_4$ for the model with expansion to $N_2$ crypts. The rest of the parameters are as in *Appendix 1—figure 5*.

Therefore, to proceed, we will fix the crypt fission rates $\gamma_3$ and $\gamma_4$ to their values in Table *Appendix 1—table 2*, and fit the model to the advanced adenoma incidence curve to find the best rate $r_1$ and crypt carrying capacities. This is a procedure similar to that in *Appendix 1—figure 6*, except type 6 crypt populations grow to a given size. The fitting error for the three cases (that is, growth to size 1, size $10^2$, and size $10^5$ crypts) is shown in *Appendix 1—figure 9(a)* as a function of $r_1$. The fitting procedure was performed by using the ODE approximation (*equation (24)*), and the best fitting parameter set then used in a Gillespie simulation ($5 \times 10^5$ simulations are used and the error bars are too small to see). *Appendix 1—figure 9(b)* shows the result of the Gillespie simulation corresponding to the best-fitting parameters for the case of expansion to $10^5$ crypts. The ODE prediction and the fitted curve for the expansion to $10^5$ crypts look identical.

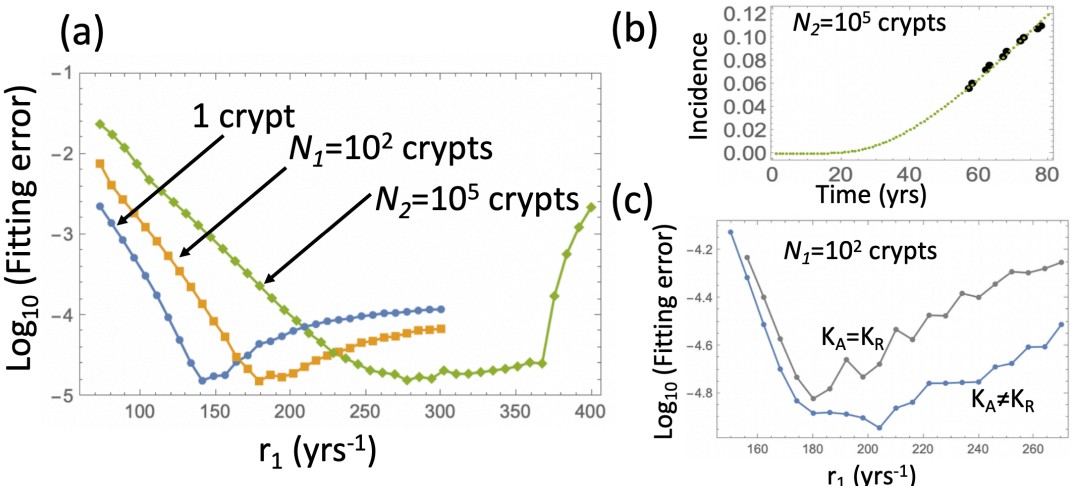

**Appendix 1—figure 9.** Fitting the adenoma incidence curve to the nonlinear model with an explicit expansion phase. Under fixed crypt fission rates, for each value of $r_1$ the best fitting value of $K_A = K_R$ was found. (**a**) The fitting error is shown as a function of $r_1$ for the original model (blue line, same as in *Appendix 1—figure 6(b)*); the model with expansion to $N_1$ crypts (yellow curve), and the model with expansion to $N_2$ crypts (green curve). (**b**) The fitted incidence curve (Gillespie simulation) for the case of expansion to $10^5$. The ODE prediction and the fitted curve for the expansion to $10^2$ crypts look identical. (**c**) The reduction in error resulting from relaxing the restriction $K_A = K_A$. The rest of the parameters are as in *Appendix 1—figure 6(a-c)*.

## Compartment dynamics and pathways to advanced adenoma

Relaxing the restriction $K_A = K_R$ improves the fitting, as shown in *Appendix 1—figure 9(c)*. We will explore compartment dynamics in this more general setting, given that by AIC, the model with carrying capacities $K_A$ and $K_R$ separately fitted is more powerful. *Appendix 1—figure 10* shows the procedure of fitting in the case where type 6 crypts grow to size $10^2$ before detection. For each value of $r_1$, the best fitting pair $(K_A, K_R)$ was determined and the resulting error plotted (panel (a)). Panel (c) shows examples of heatplots of the error as a function of $K_A$ and $K_R$ (for a fixed $r_1$ value). Depending on the size of the resulting error and the location of the minimum in the $(K_A, K_R)$ space, three groups of fits can be distinguished. Of particular interest in the group marked by orange and green points. For the fits in the "orange" group, we have $K_A > K_R$, that is, the best fitting carrying capacity that characterizes type-3 crypts is larger than that for type-4 and 5 crypts. In contrast to that, the "green" group has $K_A < K_R$. This can be seen in *Appendix 1—figure 10(b)*, where the best fitting pairs $(K_A, K_R)$ are shown for each $r_1$ value. We reject the "green" group of fits because it corresponds to a larger carrying capacity of type-4 and 5 crypts compared to that of type-3 crypts, which is inconsistent with observations; in addition,

the fitting error is minimized for one of the fits in the "orange" group. The same procedure was also performed for the case where type-6 crypts grew to size $10^5$ (see **Appendix 1—figure 11**), with very similar results.

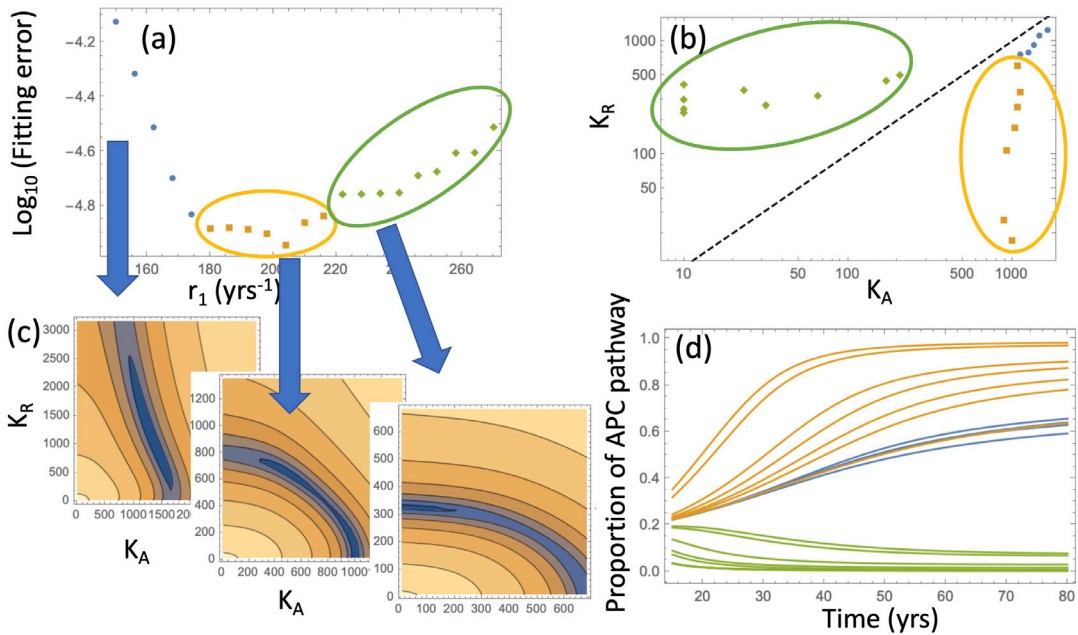

**Appendix 1—figure 10.** Pathways to adenoma: Fitting the incidence curve to the nonlinear model with an explicit expansion phase. Under fixed crypt fission rates, for each value of $r_1$ the best fitting pair $(K_A, K_R)$ was found. (**a**) The fitting error is shown as a function of $r_1$; 3 groups of parameter combinations are marked by blue, orange, and green. (**b**) The best fitting pairs $(K_A, K_R)$ for each of the points in the three groups. (**c**) Heatplots of the fitting error as a function of $(K_A, K_R)$, for 3 different values of $r_1$, one from each group. (**d**) Proportion of the APC pathway (formula **Moolgavkar, 1978**) plotted as a function of time for different values of $r_1$, using the same color code for the three groups. Expansion to $10^2$ crypts is assumed. The rest of the parameters are as in **Appendix 1—figure 6a-c**.

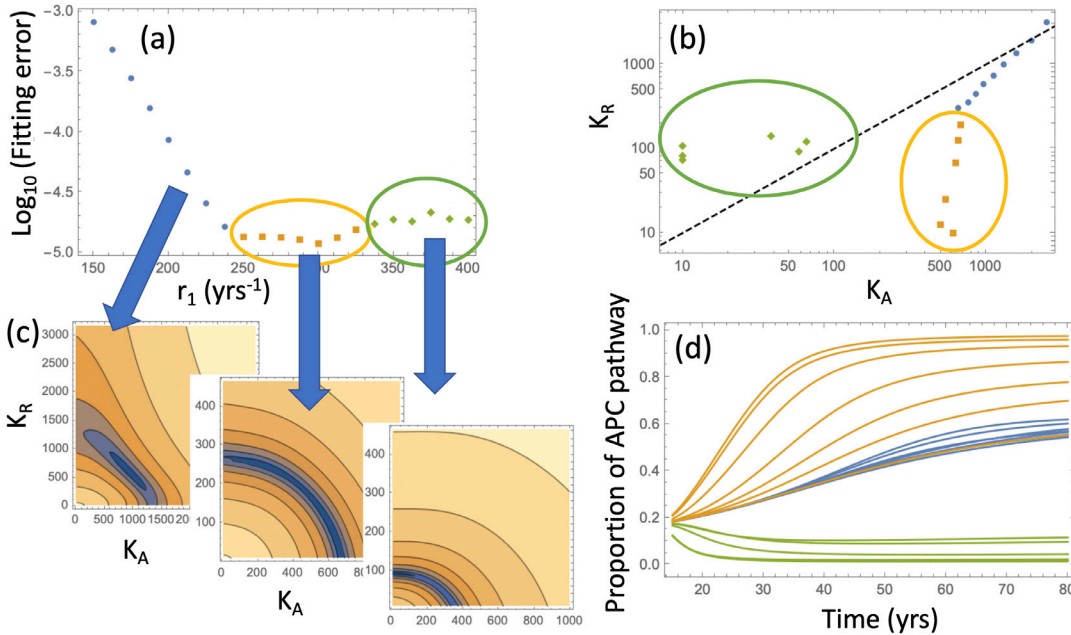

**Appendix 1—figure 11.** Same as **Appendix 1—figure 10**, except expansion to $10^5$ crypts is assumed.

In order to determine the likelihood of type $y_6$ produced from cells of type $y_3$ or cells of type $y_5$, we looked at the probabilities of the two pathways, $P_{APC}$ and $P_{KRAS}$, which stand for the probability to create adenoma by first inactivating the APC gene and then adding a gain-of-function mutation in KRAS gene, or by first activating KRAS and then inactivating the APC gene, see panel (a) of **Figure 3** of the main text. The two probabilities satisfy the following equations,

$$\dot{P}_{APC} = R_{36} n_3 (1 - P_{APC}), \quad P_{APC}(0) = 0, \tag{25}$$

$$\dot{P}_{KRAS} = R_{56} n_5 (1 - P_{KRAS}), \quad P_{KRAS}(0) = 0. \tag{26}$$

The proportion of adenomas that originated through the APC pathway is given by

$$P_{APC} / (P_{APC} + P_{KRAS}). \tag{27}$$

This function is plotted in **Appendix 1—figure 10(d)** for the parameters corresponding to both "orange" and "green" groups of fits. We can see that for parameter combinations from the orange group the APC-pathway is predominant, while for parameter combinations from the "green" group the KRAS pathway is predominant. The same qualitative results are observed in **Appendix 1—figure 11(d)**.

For further simulations, we used the two best fitting parameter combinations obtained from the fitting procedures in **Appendix 1—figure 10** and **Appendix 1—figure 11**. The parameter values obtained by means of these fitting procedure are summarized in Table **Appendix 1—table 3**.

**Appendix 1—table 3.** Parameters obtained from the fitting procedures in **Appendix 1—figure 10** and **Appendix 1—figure 11**.

| Parameter set # | $N_i$ (crypts) | $\Delta T_i$ (yrs) | $r_1$ (yrs-1) | $K_A$ | $K_R$ | Comment |
|---|---|---|---|---|---|---|
| 1 | $10^2$ | 4.79 | 204 | 1000 | 17 | best fit |
| 2 | $10^2$ | 4.79 | 198 | 1039 | 171 | 2nd best fit |
| 3 | $10^5$ | 11.97 | 300 | 607 | 10 | best fit |
| 4 | $10^5$ | 11.97 | 288 | 631 | 68 | 2nd best fit |

System (LABEL:n1non-23, 12) does not only allow fitting the model to advanced adenoma incidence data, but also shows the prediction for the dynamics of the crypts of different mutational status leading up to the advanced adenoma formation, as shown in figure **Appendix 1—figure 12(a)**. Using the best fitting parameters of **Appendix 1—figure 6(b)**, we can visualize the mean trajectories for the numbers of crypts in each compartment, $n_i$ for $i = 1, \ldots, 5$. The stochastic Gillespie simulations provide additional information on the statistics of crypt numbers. **Appendix 1—figure 12(b)** shows the numerically obtained probability distributions for the crypt numbers in each compartment, at the time when the first type 6 crypt is created. These simulations of panel (b) correspond to the same parameter set as in panel (a); both assume expansion to $10^2$ crypts and $K_A = K_R$. Panel (c) presents similar simulations but for the best fitting parameters where expansion to $10^5$ crypts takes place. The mean crypt numbers are given in blue in each histogram; we can see that the orders of magnitude are similar for both cases. Similar results are obtained when the restriction $K_A = K_A$ is dropped, see **Appendix 1—figure 13**; as expected, the size of the type-3 compartment is somewhat larger and the size of the type-5 compartment somewhat smaller than the corresponding values for the $K_A = K_R$ case. Note that **Appendix 1—figure 12** and **Appendix 1—figure 13** do not contain information on pathways to advanced adenoma. In other words, shown are just the numbers of crypts of each type that are present in the entire simulated colon, at the time when the first crypt associated with the

advanced adenoma is generated, regardless of whether it was generated by a mutation of a type 3 or a type 5 crypt.

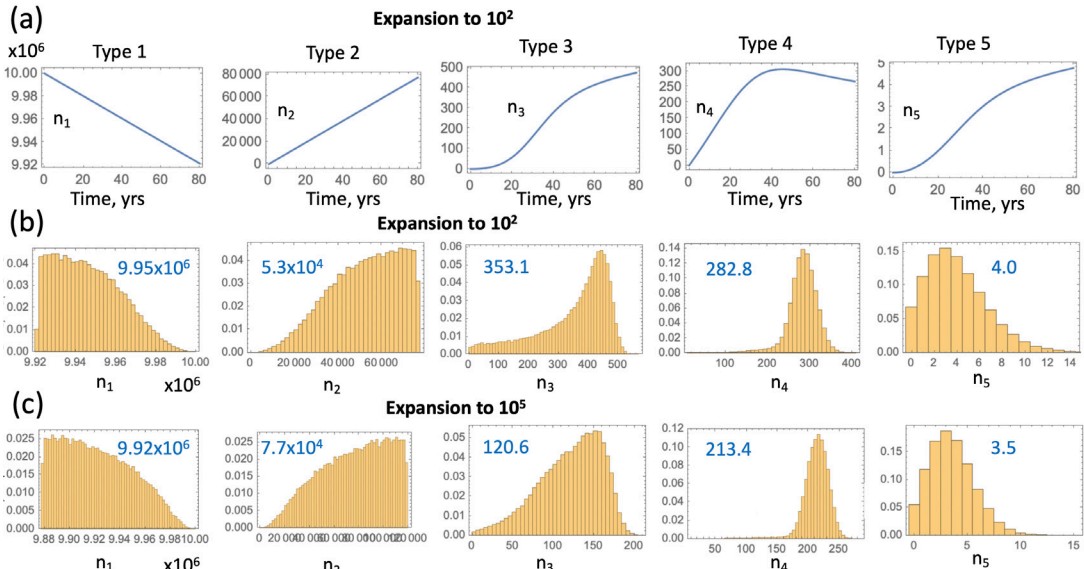

**Appendix 1—figure 12.** Compartment dynamics for the best fitting models with $K_A = K_R$. (**a**) The expected crypt numbers (using equations (LABEL:n1non-23)) for the 5 compartments are shown for the 2nd parameter set in table *Appendix 1—figure 9(c)* (expansion to $10^2$ crypts). (**b**) For the same parameter values, probability distributions for the crypt numbers are shown at the time when the first type 6 crypt is created ($5 \times 10^5$ simulations were run); mean crypt numbers are shown in blue. (**c**) Same as (**b**) but for the 3rd parameter set in table *Appendix 1—figure 9(c)* (expansion to $10^5$ crypts).

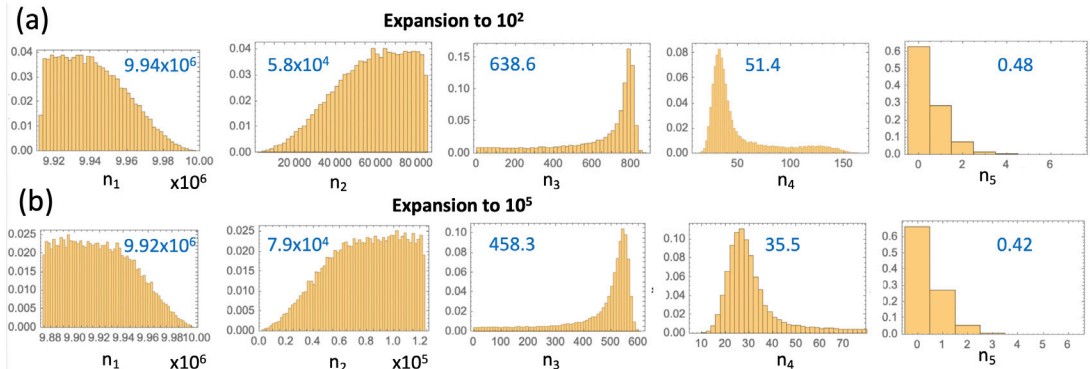

**Appendix 1—figure 13.** Same as *Appendix 1—figure 12(b-c)*, but without the restriction $K_A = K_R$. (**a**) Parameter set #2 from Table *Appendix 1—table 3* was used. (**b**) Parameter set #4 from Table *Appendix 1—table 3* was used.

## Aspirin dosage and modeling aspirin's effect on the crypt dynamics

In mouse experiments performed in *Shimura et al., 2020* we used the doses of 15 mg/kg, 50 mg/kg, and 100 mg/kg aspirin each day. Using the conversion table in *Drew et al., 2016*, and a mass of a human of 70 kg, we obtain the equivalent daily human doses of 85.1 mg/day, 283.5 mg/day, and 567 mg/day.

To compare this with the weekly intake of the participants in the study of *Chan et al., 2004*, we note that a standard aspirin tablet contains 325 mg, such that 2 tablets a day (i.e. 14 tablets per week) is 650 mg/day. The patients were grouped by their intake into 0.51.5 tablets per week, 25 tablets per week, 614 tablets per week, and $gt_{14}$ tablets per week. Therefore, the highest dose

administered in our mouse experiments (100 mg/kg aspirin per day) most closely matches the intake of the 614 tablets per week group, see *Table 1* of the main text.

To explore the effect of aspirin on the incidence of advanced adenoma, we used Gillespie simulations of system (LABEL:n1non-23), also including a growth phase of type 6, $\dot{n}_6 = R_{36}n_3 + R_{56}n_5 + (\gamma_6 - \delta_6)n_6$.

In these simulations, aspirin-free parameter values were used for $0 \leq t < T_{start}$, and then during an interval of aspirin treatment $T_{start} \leq t \leq T_{end}$, parameter values modified by aspirin were used, see *Figure 4(b)* of the main text. As before, simulations were stopped at $t = 80$ yrs or when the target number of type 6 crypts were generated, whichever happened first. Incidence curves and relative risk were constructed by processing a large number of such simulations. $2.5 \times 10^5$ independent simulations per condition were used unless otherwise noted. The error bars for the age incidence curves or the relative risk are too small to be seen.

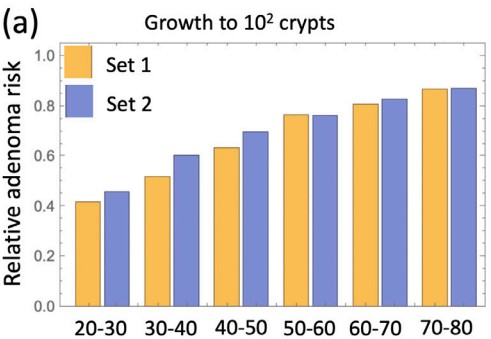 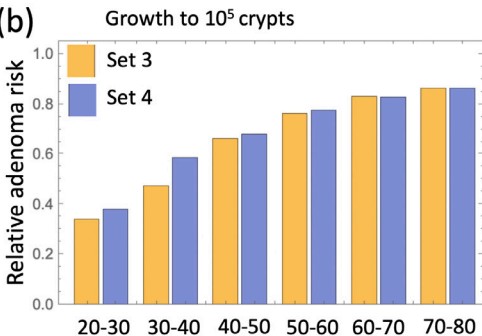

**Appendix 1—figure 14.** extends the results of main text *Figure 4(a)* and shows that relative adenoma risk predicted for each decade of treatment is very similar for all parameter sets (#1-4, Table *Appendix 1—table 3*).

Figure *Appendix 1—figure 14*: Relative incidence of advanced adenoma, where (a) growth to $10^2$ type 6 crypts is assumed and parameter sets #1 and #2 (Table *Appendix 1—table 3*) are compared, and (b) growth to $10^5$ type 6 crypts is assumed and parameter sets #3 and #4 are compared. Aspirin affects type 2–6 cells, both through conversion rates and crypt fission/death. Treatment is applied for different decades (as marked under the pairs of bars), and the relative risk is evaluated at the end of the treatment decade. $2.5 \times 10^5$ independent simulations are used for each condition.

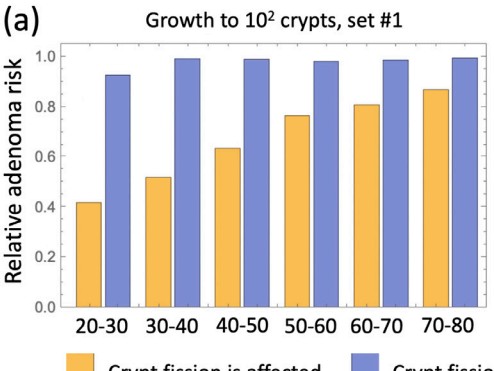 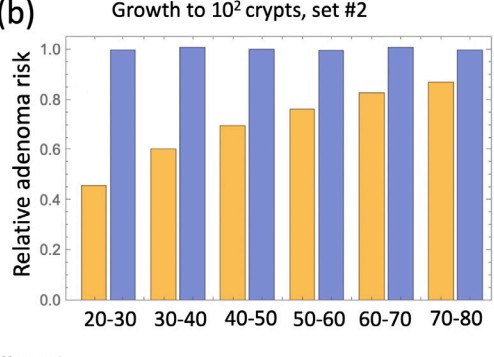

**Appendix 1—figure 15.** Relative incidence of advanced adenoma, where (**a**) parameter set #1 (Table *Appendix 1—table 3*) and (**b**) parameter sets #2 is used. In each panel, a comparison is presented between the case where aspirin affects crypt fission/death rates (yellow) and where it does not affect crypt fission/death rates (blue). Aspirin affects cellular kinetics through conversion rates in all cases; types 2-6 are affected. $2.5 \times 10^5$ independent simulations are used for each condition.

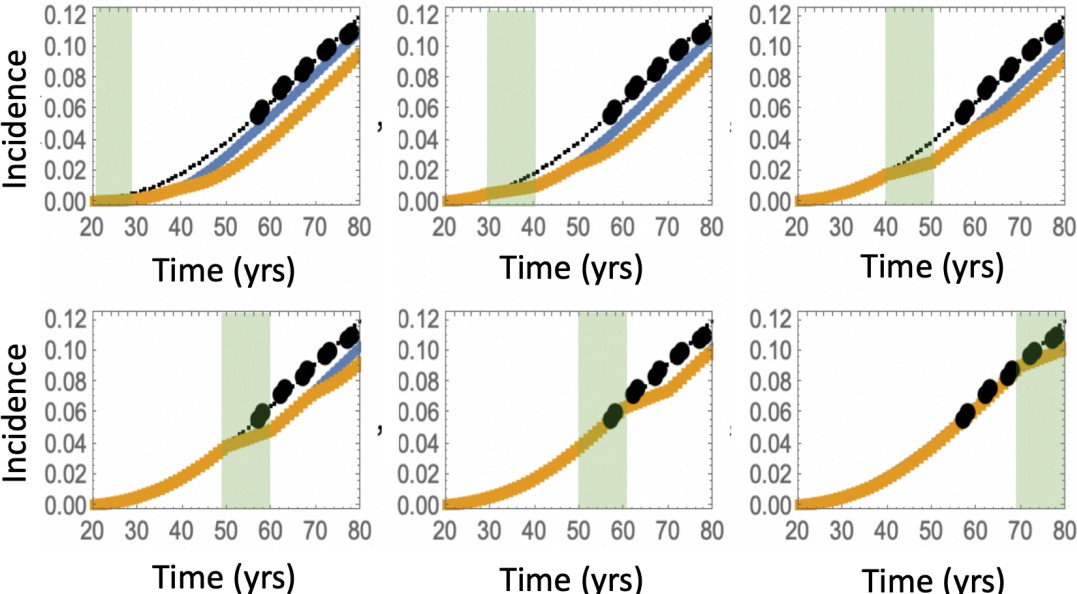

**Appendix 1—figure 16.** Predicted advanced adenoma incidence in the absence of aspirin treatment (thin black lines are the fitted curves and black dots are incidence data); under aspirin treatment where the drug affects types 2–6 (yellow lines), under aspirin treatment where the drug affects type 6 only (blue lines). Each panel corresponds to aspirin treatment administered during one decade (20-30 years, 30–40 years, etc). The treatment period is shaded light green. It was assumed that type 6 crypts grow to $10^5$, and set #4 in Table **Appendix 1—table 3** was used. $2.5 \times 10^5$ independent simulations are used for each condition.

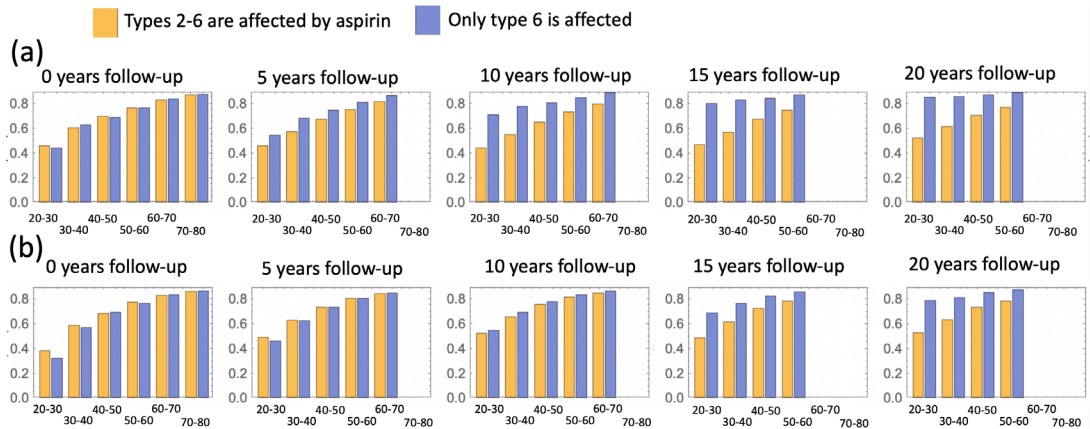

**Appendix 1—figure 17.** A comparison of predicted relative advanced adenoma risk under the assumption that cell types 2–6 (yellow bars) and only type 6 cells (blue bars) are affected by aspirin. Aspirin is administered during different decades of patients' life (as marked under each pair of bars), and different panels correspond to different durations of the follow-up period. (**a**) Parameter set #2, and (**b**) parameter set #4 (Table **Appendix 1—table 3**) is used; $2.5 \times 10^5$ independent simulations are used for each condition.

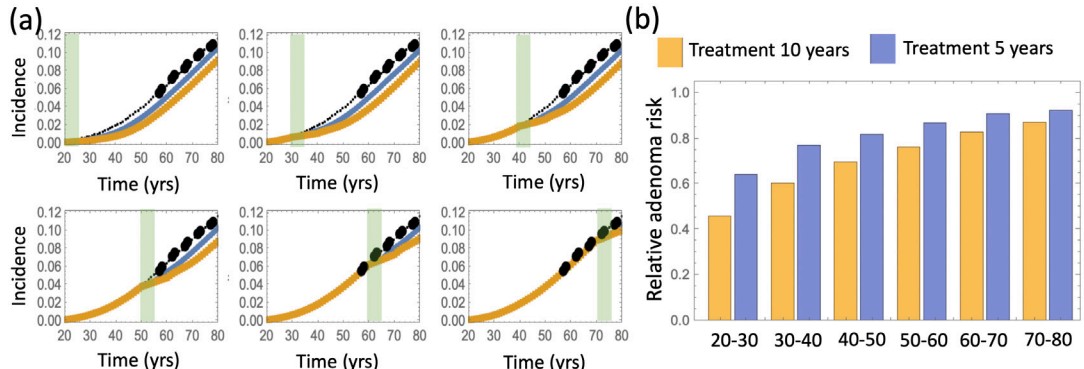

**Appendix 1—figure 18.** A comparison of predicted relative advanced adenoma risk under the assumption that aspirin treatment lasts 10 years (yellow) or 5 years (blue). (**a**) Predicted advanced adenoma incidence in the absence of aspirin treatment (thin black lines are the fitted curves and black dots are incidence data); the incidence curve fo patients treated for 10 (yellow) and 5 (blue) years. Aspirin is administered during different decades of patients' life (the shorter, 5 year intervals, are marked green). (**b**) A comparison of predicted relative advanced adenoma risk under the assumption of treatment for 10 years (yellow bars) and 5 years (blue bars). All cell types 2–6 are affected by aspirin (both through conversion and fission/death). Parameter set #2 is used; $2.5 \times 10^5$ independent simulations are used for each condition.

*Appendix 1—figure 15* explores the individual contributions of the two possible mechanisms by which the effect of aspirin could be manifested. The yellow bars correspond to the effect on both conversion rates and crypt fission/death rates, and the blue bars only include the effect on conversion rates. It is clear that the effect on conversion rates is not as strong compared to the effect on the crypt fission/death rates.

*Appendix 1—figure 16* is similar to main text *Figure 4(a)*, except is contains additional incidence functions that correspond to aspirin only affecting type 6 cells (blue lines). Not surprisingly, the resulting change in adenoma risk is smaller, that is, the blue line (effect on type 6 cells only) is closer to the thin black line (no-aspirin adenoma incidence) than the yellow line (where aspirin affects all types 2–6). It is interesting that the difference becomes larger with time. In *Appendix 1—figure 17* we change the length of the follow-up (see *Figure 4(b)* of the main text), and the difference between the two assumptions (crypts 2–6 affected vs only type 6 crypts are affected) increases as time goes by.

*Appendix 1—figure 18* shows the effect of a shorter aspirin duration time

