## [Editor Report]

This work develops a multistage/component mathematical model to analyze advanced colorectal adenomas and the impact that aspirin therapy has on adenoma formation rates. This study will be interesting to the cancer evolution community and in particular those interested in colorectal cancer incidence. While the model is mainly focused on aspirin chemoprevention, the model could be adapted to test other putative preventative agents, and thus could have a broad impact.

---

## [Decision Letter]

**Decision letter after peer review:**

Thank you for submitting your article "The protective effect of aspirin in colorectal carcinogenesis: a multiscale computational study from mutant evolution to age incidence" for consideration by *eLife*. Your article has been reviewed by 3 peer reviewers, and the evaluation has been overseen by a Reviewing Editor and a Senior Editor. The following individuals involved in review of your submission have agreed to reveal their identity: E Georg Luebeck (Reviewer #2); Andrew Chan (Reviewer #3).

As is customary in *eLife*, the reviewers have discussed their critiques with one another. What follows below is the Reviewing Editor's edited compilation of the essential and ancillary points provided by reviewers in their critiques and in their interaction post-review. Please submit a revised version that addresses these concerns directly. Although we expect that you will address these comments in your response letter, we also need to see the corresponding revision clearly marked in the text of the manuscript. Some of the reviewers' comments may seem to be simple queries or challenges that do not prompt revisions to the text. Please keep in mind, however, that readers may have the same perspective as the reviewers. Therefore, it is essential that you attempt to amend or expand the text to clarify the narrative accordingly.

Essential revisions:

1)) Overall, the manuscript would benefit from a more precise explanation of the assumptions used in the models presented. This would include a more clear discussion/rationalization of advanced adenoma, adenoma classification, and how aspirins effect was implemented at the crypt level (see first reviewers comments)

2) There needs to be increased justification (or modification of the model) for why the assumption of zero crypt death/fusion.

3) Several reviewers mention limitations/concerns with the reliance on mutant KRAS (i.e. lack of determining KRAS status in adenomas/cancers, lack of APC/KRAS mutational status in predicting aspirin response and that the effect estimates are based on preclinical work using relatively high doses of aspirin. Please address these concerns in the manuscript and in response.

4) Addressing the concern from reviewer #3 regarding the assumption that the cell of origin for CRC is an ISC as opposed to more recent theories suggesting alternative origins and suggestions on expanding the discussion to include more recent literature on age-differences in aspirins effects on CRC.

*Reviewer #1 (Recommendations for the authors):*

In addition to the points raised in the public review, I have the following comments:

It is unclear to me why the effect of reducing the fitness of type 6 is done through reducing the rates of mutation to type 6 (R36 and R56). What is the justification for that?

For the scenario when aspirin reduces gamma3 and gamma4, why does it also not reduce gamma5?

It would also be important to discuss more precisely (i.e. by referring to specific mathematical models) how the findings that aspirin changes division and death rates in cell culture, where there is no tissue hierarchy, translates to the in vivo setting, where the effects of aspirin may be felt by crypts, stem cells or progenitor cells.

*Reviewer #3 (Recommendations for the authors):*

I read the manuscript with great interest and was pleased to see that the authors took care to acknowledge the limitations and clearly explain the base assumptions used in their approach. The manuscript is well-written and there are only minimal additions that may improve the manuscript.

– Unless I have misunderstood, the basic model allows us to understand the probability of developing APC and/or KRAS mutant adenomas (and using this as a relative measure of the 'advanced' nature of the in silico 'neoplasm'). I appreciated the discussion relative to the formation of aberrant crypts vs. adenomas vs. more advanced precancers. However, it seems that everything operates on the basic premise that an intestinal stem cell must be the tumor cell of origin and that aspirin is having specific effects on these cells. This assumption may be too much an oversimplification to allow the model to have broad reaching applicability. For example, recent work has begun to describe that the tumor cell of origin may not be the classical intestinal stem cell in all CRC cases, especially with advancing age or under different dietary stressors, and separately in parallel, that aspirin may have effects on cell differentiation/states (e.g. Devall et al. Cancer Prev Res 2021), mechanism may be cell-context specific, or be significantly impacted by epithelial cell extrinsic factors not included in the model (e.g. gut microbiome, see A. Prizment et al. Aliment Pharmacol Ther 2020; C. Brennan et al. mBio 2020; R Zhao et al. Gastro 2020) While the authors do describe that these assumptions may be limiting, I think prudent for the authors to discuss the specific impacts of the assumption that aspirin has a direct effect on intestinal stem cells being tumor cell of origin has on model interpretation. How would estimates be potentially influenced if intestinal stem cells were not the target cell or aspirin only had effects in specific cell types or by cell extrinsic factors? What do these assumptions have on the broader generalizability of this model? Can the authors expand on how this may be expected to be accounted for in the future? What additional information or type of data is needed from clinical and preclinical experiments to allow for more accurate biological modeling of these complex interplays? The last question is particularly important to understand the broader impact of these findings and if the models have to potential to more directly inform future research.

– Similarly, I think more discussion could be owed to the emerging literature around the intersection of age-differences and aspirin mechanism, especially in light of the recent results from the ASPREE trial that described an increase in cancer death as a result of aspirin intervention among adults over age 70. Is it possible to model the timing of aspirin intervention using this model, particularly in view of the probability differences arising from differential mutational priming outcomes (APC-/- vs. APC -/+ vs. KRAS+, etc.). The ASPREE results demonstrated that the increase in cancer mortality was not driven by a change in cancer incidence and I wonder if the authors can try to model these effects or at least discuss how the model findings should be interpreted in view of these recent results from trials.

– The manuscript primarily discusses the CAPP2 trial as the evidence supporting aspirin chemoprevention of colorectal cancers. Although this obviously has clear implications for placing the results in the context of prevention of CRCs in Lynch syndrome, these tumors are neither sporadic, nor arise via the pathways included in the model. The authors could broaden the background to include the preponderance of evidence for the preventive effects in sporadic cases (or even FAP patients which are known to have APC mutant cancers) where aspirin has had less of a potent chemopreventive effect than in Lynch syndrome. However, these data are relevant to the most extreme phenotype in their model.

[Editors' note: further revisions were suggested prior to acceptance, as described below.]

Thank you for resubmitting your work entitled "Aspirin's effect on kinetic parameters of cells contributes to its role in reducing the incidence of advanced colorectal adenomas, shown by a multiscale computational study" for further consideration by *eLife*. Your revised article has been evaluated by the reviewers and by the Editors.

The manuscript has been improved but there are some remaining issues that need to be addressed, as outlined below:

*Reviewer #1 (Recommendations for the authors):*

The authors have carefully addressed previous concerns, and I am satisfied with the revision.

*Reviewer #2 (Recommendations for the authors):*

The revised manuscript has gained considerably in strength and, by in large, clarifies the main points raised by the reviewers. I appreciate the extra work that went into refining/extending the model analysis, in particular the addition of a growth phase for the type 6 (advanced) adenoma.

Two lingering points. I hope they can be addressed.

1. Please clarify whether the size distribution provided for type 3 (APC-/-) adenoma in Figure Appendix 1 Figure 12 and 13 refer to the particular type 3 adenoma (clone) in which the advanced adenoma first developed or to the entire population of type 3 adenomas in the colon. I think the authors should point out that their ODE model does NOT distinguish individual clones of abnormal crypts (ie individual adenomas). This is a limitation since detectable adenoma number of any kind (other than hyperplastics) is an important clinical factor.

2. I understand the authors' point about postulating other potential gain-of-function mutations, similar to KRAS, such as BRAF. However, BRAF is a poor example as it is associated strongly with mismatch repair deficiency and SSAs, leading frequently to hypermutated cancers. While there may be other yet unidentified gain-of-function drivers for the advanced adenoma, there may also simply none required given the epigenetic plasticity and adaptive epigenetic changes as adenomas sojourn for years. In any case, the mention of BRAF is somewhat misleading in the context defined by the authors.

*Reviewer #3 (Recommendations for the authors):*

Thank you for a very complete response. The manuscript is excellent!

---

## [Author Response]

Essential revisions:1)) Overall, the manuscript would benefit from a more precise explanation of the assumptions used in the models presented. This would include a more clear discussion/rationalization of advanced adenoma, adenoma classification, and how aspirins effect was implemented at the crypt level (see first reviewers comments)

Following the referees’ insightful comments, we have rewritten and extended the parts of the manuscript that describe the model, both in the main text and in the Appendix. The models and the underlying assumptions are now described in greater detail both in the main text and in Materials and methods. This includes a discussion about adenoma classification, and the nature of the mutations. We have significantly expended Section 2 of Appendix 1 that describes all the model parameters. The part of the Results section in the main text that describes the effect of aspirin is now significantly rewritten (starting p13), including the new Figure 4 and Table 1, and we have also added a new Section 6 in Appendix 1, where further details are provided. In particular, we distinguish the effect that aspirin may have on the intra- and inter-crypt dynamics (that is, its role in modifying conversion rates and fission/crypt death rates, respectively). We have now explicitly included three different doses of aspirin, see new Table 1 for aspirin dosage in mice, the human equivalent, and the resulting fold differences in the division and death rates of cells that were previously measured by us and implemented in this study. The timing of aspirin administration is explained in the schematic of the new Figure 4(b) of the main text.

2) There needs to be increased justification (or modification of the model) for why the assumption of zero crypt death/fusion.

Spontaneous crypt loss is part of the model; it is incorporated in the crypt death rate, d, in the system of equations presented in the main text (as well as system (19-23) in Appendix 1). All the simulations in the main text (e.g. Figures2-4) include a nonzero crypt death rate. We have also researched the effect of crypt death rate, by performing simulations with both zero and non-zero values of d (see Appendix 1- Figure 6). Since the new version of the model also takes into account the expansion phase of type 6 crypts, a nonzero crypt death rate associated with type 6 crypts is also included. It plays a role in the stochastic model, as sometimes, a newly generated crypt may spontaneously disappear. It is also assumed to be affected by aspirin, when we consider the inter-crypt dynamics.

3) Several reviewers mention limitations/concerns with the reliance on mutant KRAS (i.e. lack of determining KRAS status in adenomas/cancers, lack of APC/KRAS mutational status in predicting aspirin response and that the effect estimates are based on preclinical work using relatively high doses of aspirin. Please address these concerns in the manuscript and in response.

We now discuss these issues at length. As pointed out by the reviewers, it has been reported that among non-hypermutated colorectal tumors, KRAS was mutated in only about 43% of patient samples, indicating the importance of alternative evolutionary pathways. Our model, however, does not depend on the identity of particular mutations, but assumes the occurrence of mutation types, which are the inactivation of a tumor suppressor gene (which is a loss-of-function mutation, e.g. APC-/-), and a gain-of-function mutation, which can be in KRAS or an alternative gene, such as BRAF. Our model predictions hold as long as the evolutionary pathway to advanced adenomas involves a loss-of-function mutation and a gain of function mutation, regardless of their identity.

Regarding aspiring dose, we now provide a table relating doses used in our in vivo experiments to the number of aspirin pills used in reference [14]. Further, we now show simulation results for a strong, intermediate, and light aspirin dose, based on the different dosing regimes in our experiments, and relate our predictions to the epidemiological observations in reference [14]. This is described in detail in the Result section, and the accompanying graphs are shown in Figure 4 (as well as Section 6 of Appendix 1).

4) Addressing the concern from reviewer #3 regarding the assumption that the cell of origin for CRC is an ISC as opposed to more recent theories suggesting alternative origins and suggestions on expanding the discussion to include more recent literature on age-differences in aspirins effects on CRC.

We have now addressed these issues in detail. Regarding the cell of origin issue, the Discussion section now includes the following: “While we concentrated our model description around stem cells as the cell of origin that drives disease, the model defines this population as having the ability to self-renew thus maintaining the expansion of the tumor. Hence, this cell population in the model could also correspond to compartments downstream in the differentiation pathway, such as transit amplifying cells, given the marked plasticity within the intestinal epithelium. The model is thus in principle consistent with hypotheses that colorectal cancer might have a different cell of origin.”

Regarding the effect of aspirin in older ages, we now provide an extensive discussion of the ASPREE trial (second to last paragraph, Discussion section), and describe how these observations can be interpreted in the light of evolutionary models.

Reviewer #1 (Recommendations for the authors):In addition to the points raised in the public review, I have the following comments:It is unclear to me why the effect of reducing the fitness of type 6 is done through reducing the rates of mutation to type 6 (R36 and R56). What is the justification for that?

We thank the referee for pointing this out. This statement was not precise. Changes in the relative fitness of different types (including type 6) are implemented by modifying the parameters r_i_ (relative fitness of cells), and this in turn changes (in a non-linear fashion) the relevant conversion rates. This is now explained in the new Section 2 of Appendix 1, and also in the heavily revised part of the main text (Results) where we describe modeling the effects of aspirin (see also the new Section 6 of Appendix 1 for details).

For the scenario when aspirin reduces gamma3 and gamma4, why does it also not reduce gamma5?

In our model, we assume that gamma5=gamma4, and therefore both are affected in the model. This was unclear in the original version and is now stated explicitly.

It would also be important to discuss more precisely (i.e. by referring to specific mathematical models) how the findings that aspirin changes division and death rates in cell culture, where there is no tissue hierarchy, translates to the in vivo setting, where the effects of aspirin may be felt by crypts, stem cells or progenitor cells.

We have now done this in the revised manuscript.

Regarding stem cells: There are data in the literature showing that stem cells in particular are also affected by aspirin, which we cite. To describe how the effect on stem cells was implemented, we included the following text: “For the purposes of our model, it is the combined effect of aspirin on cell division and death rates that changes the cells' relative fitness and decreases the probability of crypt conversion. To translate this information into the fold decrease in SC fitness, we note that, while the fold-reduction in division rate could be directly implemented, an increase in death rate is less straightforward. This is because in contrast to cell lines, with SCs, cell removal can occur through a combination of apoptosis and loss through differentiation, which might be the dominant component in the colorectal tissue. Therefore, if the rate of SC apoptosis is increased, say, two-fold in the presence of aspirin, this does not translate to a two-fold reduction in SC fitness. In the extreme scenario of zero SC death in the absence of aspirin, a two-fold increase in this parameter will not lead to a change in SC fitness. To calculate the fitness factor, we assumed that the removal rate of SCs, *d*, is comprised of 90% differentiation and 10% apoptosis, and that it is the latter that is affected by aspirin. If in the absence of aspirin, cellular fitness is given by the ratio r/d, then in the presence of aspirin this changes to r/d x F_r_/(0.9+0.1F_d_), which gives the fitness factor in Table 1. This factor enters into the crypt conversion rate, see Section 2 of Appendix 1. In particular, if only type 6 is affected, then rates R_36_ and R_56_ will experience a reduction. If types 2-6 are affected, then all conversion rates will be reduced.”

Regarding inter-crypt dynamics, the following text was included: “In addition to affecting cellular fitness within the crypts, it is also logical to assume that aspirin reduces crypt fission rates and increases crypt death rates (the bottom row of the table in Figure 4(a)). This is supported by data [57], and the rationale behind this assumption is that crypt fission is ultimately connected with divisions of individual cells, and crypt death is associated with cell death. Therefore, we assume that under aspirin treatment, γ_i_ →F_r_ γ_i_ and δ→Fdδ (that is, the fold-differences apply to the crypt fission and death rates). Again, this could affect the most modified crypts only (type 6), thus reducing the rate g_6_ and increasing the death rate d _6_; alternatively, this could affect to all type 2-6 crypts, thus reducing all the crypt fission rates and increasing all the crypt death rates.“

Regarding inter-crypt dynamics, the following text was included: “In addition to affecting cellular fitness within the crypts, it is also logical to assume that aspirin reduces crypt fission rates and increases crypt death rates (the bottom row of the table in Figure 4(a)). This is supported by data [57], and the rationale behind this assumption is that crypt fission is ultimately connected with divisions of individual cells, and crypt death is associated with cell death. Therefore, we assume that under aspirin treatment, γ_i_ →F_r_ γ_i_ and δ→Fdδ (that is, the fold-differences apply to the crypt fission and death rates). Again, this could affect the most modified crypts only (type 6), thus reducing the rate g_6_ and increasing the death rate d _6_; alternatively, this could affect to all type 2-6 crypts, thus reducing all the crypt fission rates and increasing all the crypt death rates.“

Reviewer #3 (Recommendations for the authors):I read the manuscript with great interest and was pleased to see that the authors took care to acknowledge the limitations and clearly explain the base assumptions used in their approach. The manuscript is well-written and there are only minimal additions that may improve the manuscript.– Unless I have misunderstood, the basic model allows us to understand the probability of developing APC and/or KRAS mutant adenomas (and using this as a relative measure of the 'advanced' nature of the in silico 'neoplasm'). I appreciated the discussion relative to the formation of aberrant crypts vs. adenomas vs. more advanced precancers. However, it seems that everything operates on the basic premise that an intestinal stem cell must be the tumor cell of origin and that aspirin is having specific effects on these cells. This assumption may be too much an oversimplification to allow the model to have broad reaching applicability. For example, recent work has begun to describe that the tumor cell of origin may not be the classical intestinal stem cell in all CRC cases, especially with advancing age or under different dietary stressors, and separately in parallel, that aspirin may have effects on cell differentiation/states (e.g. Devall et al. Cancer Prev Res 2021), mechanism may be cell-context specific, or be significantly impacted by epithelial cell extrinsic factors not included in the model (e.g. gut microbiome, see A. Prizment et al. Aliment Pharmacol Ther 2020; C. Brennan et al. mBio 2020; R Zhao et al. Gastro 2020) While the authors do describe that these assumptions may be limiting, I think prudent for the authors to discuss the specific impacts of the assumption that aspirin has a direct effect on intestinal stem cells being tumor cell of origin has on model interpretation. How would estimates be potentially influenced if intestinal stem cells were not the target cell or aspirin only had effects in specific cell types or by cell extrinsic factors? What do these assumptions have on the broader generalizability of this model? Can the authors expand on how this may be expected to be accounted for in the future? What additional information or type of data is needed from clinical and preclinical experiments to allow for more accurate biological modeling of these complex interplays? The last question is particularly important to understand the broader impact of these findings and if the models have to potential to more directly inform future research.

These are important points to address in the manuscript. Regarding the cell or origin issue, we did write the text under the premise that colorectal cancer stem cells are driving tumor development and growth, due to the large emphasis on this in the literature. The model, however, is more general in that it assumes a dividing population of cells that initiates and drives tumor growth, without specifying their identity. This could correspond not only to stem cells, but also to cell populations further downstream in the differentiation pathway (such as transit amplifying cells), and could include initial mutant acquisition, followed by dedifferentiation. In the revised manuscript, we now take this broader perspective rather than concentrating solely on stem cells. This is provided in the Discussion section as follows: “Another uncertainty concerns the cell type in which the tumor originates, and the exact identity of the cell compartment that maintains tumor growth. While we concentrated our model description around stem cells as the cell of origin that drives disease, the model defines this population as having the ability to self-renew thus maintaining the expansion of the tumor. Hence, this cell population in the model could also correspond to compartments downstream in the differentiation pathway, such as transit amplifying cells, given the marked plasticity within the intestinal epithelium. The model is thus in principle consistent with hypotheses that colorectal cancer might have a different cell of origin [69].”

Regarding the effects of other forces driving the response to aspirin in addition to the dynamics described in our paper, we agree. In the original manuscript, we stated that although our measured aspirin-induced changes in cell kinetics can in principle account for the epidemiologically observed protective effect, it is likely that other forces, not included in the model, are also important. We mentioned the reduction of an inflammatory micro-environment by aspirin as an example. As the reviewer points out, the colorectal microbiome can also determine the level of protection provided by aspirin, and we have now added a discussion of this. This also points to the importance of quantifying the effect of the microbiome on cellular growth kinetics with and without aspirin in future work, using a combination of experimental and mathematical approaches. This would allow the inclusion of this additional complexity to our model of colorectal carcinogenesis. In the revised manuscript, this has been briefly pointed out in the Introduction, and is brought up more in depth in the Discussion sections by adding the following text:

“Moreover, other microenvironmental factors, such as the composition of the colorectal microbiome, have been shown to influence the ability of aspirin to reduce tumor growth [16-18]. This is therefore also likely to play a role in explaining the epidemiological data. Quantification of these further complexities in future work will allow us to introduce these additional aspects into the modeling framework, which would result in a refinement of predictions.”

– Similarly, I think more discussion could be owed to the emerging literature around the intersection of age-differences and aspirin mechanism, especially in light of the recent results from the ASPREE trial that described an increase in cancer death as a result of aspirin intervention among adults over age 70. Is it possible to model the timing of aspirin intervention using this model, particularly in view of the probability differences arising from differential mutational priming outcomes (APC-/- vs. APC -/+ vs. KRAS+, etc.). The ASPREE results demonstrated that the increase in cancer mortality was not driven by a change in cancer incidence and I wonder if the authors can try to model these effects or at least discuss how the model findings should be interpreted in view of these recent results from trials.

It is indeed very interesting to discuss the results of the ASPREE trial in the context of our modeling efforts, and in the more general context of evolutionary modeling. (i) In the revised manuscript, we expanded our analysis to show that the effect of aspirin on the predicted cancer incidence diminishes with the age at which aspirin treatment is initiated, being lowest if treatment is initiated at 70 years or older. This fits well with the lack of an effect of aspirin on cancer incidence in the ASPREE trial. (ii) While our modeling approach cannot make predictions about mortality (because we model evolution only up to the advanced adenoma stage), our previously published work on evolutionary dynamics might offer insights into the reasons for the observed increased mortality seen in the APSREE trial. These aspects are now discussed by adding the following paragraph to the end of the Discussion section:

“Finally, it is interesting to discuss the results of the ASPREE trial [71,72] in the context of the work presented here. This trial investigated the effect of aspirin treatment in a cohort of older individuals, 70 years or older without cardiovascular disease, dementia, or disability. It was found that cancer incidence was not significantly changed by aspirin, but that the aspirin-treated group experienced a higher rate of cancer-induced mortality. The absence of a significant effect of aspirin on cancer incidence in this study is consistent with our model predictions. Our mathematical analysis demonstrated that the effect of aspirin treatment on cancer incidence diminished when treatment was initiated in older ages. Our modeling approach, however, cannot make predictions about cancer-induced mortality, because it describes the evolutionary process up to the stage of advanced adenoma only. Our previous work [21], however, offers an interpretation of these data. Because of their advanced age, it is likely that a certain fraction of the ASPREE participants already harbored tumors that had not been detected yet due to the absence of overt clinical symptoms. In fact, a previous history of cancer was not an exclusion criterium in the trial. As the established tumors continue to grow during aspirin treatment, they likely do so with altered kinetics (reduced division rates and increased death rates, leading to a higher turnover). This means that by the time the tumor has reached a given size (e.g. at which it becomes clinically detectable), it will have undergone more cell divisions under aspirin treatment compared to the placebo group. Hence, the tumor will on average have accumulated more mutations once this detectable tumor size is reached. This in turn means that the aspirintreated tumor might be more virulent and less responsive to therapies, resulting in more deaths. The theoretically derived notion that upon detection, an aspirin-treated tumor is more evolved than a tumor that grows without aspirin [21] is supported by the ASPREE analysis, which found that aspirin-treated patients were more likely to have metastasized cancers and stage 4 cancers compared to the placebo group [71,72].“

– The manuscript primarily discusses the CAPP2 trial as the evidence supporting aspirin chemoprevention of colorectal cancers. Although this obviously has clear implications for placing the results in the context of prevention of CRCs in Lynch syndrome, these tumors are neither sporadic, nor arise via the pathways included in the model. The authors could broaden the background to include the preponderance of evidence for the preventive effects in sporadic cases (or even FAP patients which are known to have APC mutant cancers) where aspirin has had less of a potent chemopreventive effect than in Lynch syndrome. However, these data are relevant to the most extreme phenotype in their model.

We agree with this point and have broadened the background accordingly in the introduction.

[Editors' note: further revisions were suggested prior to acceptance, as described below.]

The manuscript has been improved but there are some remaining issues that need to be addressed, as outlined below:Reviewer #2 (Recommendations for the authors):The revised manuscript has gained considerably in strength and, by in large, clarifies the main points raised by the reviewers. I appreciate the extra work that went into refining/extending the model analysis, in particular the addition of a growth phase for the type 6 (advanced) adenoma.

Thank you for this positive assessment of our revisions.

Two lingering points. I hope they can be addressed.1. Please clarify whether the size distribution provided for type 3 (APC-/-) adenoma in Figure Appendix 1 Figure 12 and 13 refer to the particular type 3 adenoma (clone) in which the advanced adenoma first developed or to the entire population of type 3 adenomas in the colon. I think the authors should point out that their ODE model does NOT distinguish individual clones of abnormal crypts (ie individual adenomas). This is a limitation since detectable adenoma number of any kind (other than hyperplastics) is an important clinical factor.

Figures 12 and 13 of Appendix 1 show probability distributions of the number of crypts of each type (1 through 5) obtained in a fully-stochastic model, at the time when he first type 6 crypt is detected. This figure does not contain information on pathways to advanced adenoma. In other words, presented are just the numbers of crypts of each type that are present in the entire simulated colon, at the time when the first crypt associated with the advanced adenoma is generated, regardless of whether it was generated by a mutation of a type 3 or a type 5 crypt. We have included these clarifications in the revised text of the manuscript (Appendix 1, Section 5.3).

We also pointed out that our ODE model does not keep track of the clonality of abnormal crypts. For example, if type 5 crypt is created multiple times (by conversion) in the system, the variable n_5_ simply gives the total number of type-5 crypts. In a stochastic Gillespie model, however, it is possible to keep track of different clones by designating each newly generated crypt as a different “sub-type”, which can then clonally expand through crypt fission. This however goes beyond the scope of the current study. Text has been added to explain these points (Appendix 1, Section 5.2).

2. I understand the authors' point about postulating other potential gain-of-function mutations, similar to KRAS, such as BRAF. However, BRAF is a poor example as it is associated strongly with mismatch repair deficiency and SSAs, leading frequently to hypermutated cancers. While there may be other yet unidentified gain-of-function drivers for the advanced adenoma, there may also simply none required given the epigenetic plasticity and adaptive epigenetic changes as adenomas sojourn for years. In any case, the mention of BRAF is somewhat misleading in the context defined by the authors.

We agree. We have taken out the reference to BRAF. Further, we stated more explicitly the types of genetic events that are assumed in our model, in more general terms than in the previous version of the manuscript. That is, the model assumes that the pathway to advanced adenoma involves the inactivation of a tumor suppressor gene (e.g. APC), and a gain of function mutation (which can be KRAS or potentially an alternative). This pathway assumed by our model is supported by data available so far. We further pointed out that the model does not hold for evolutionary pathways that deviate from these assumptions. This is expressed in the following text in the manuscript (page 7, first paragraph):

“Our model, however, does not depend on the identity of particular mutations, but assumes the occurrence of mutation types; these are the inactivation of a tumor suppressor gene (which is a loss-of-function mutation, e.g. APC/-), and a gain-of-function mutation, which can be in KRAS or an alternative gene. Our model predictions hold as long as the evolutionary pathway to advanced adenomas involves these two types of mutational events, regardless of their identity. We note that our model does not apply to potential cases of advanced adenomas that might develop via pathways characterized by a different number or different types of initiating events.”